# Highly ionic-dispersed oxygen electrode for reversible proton ceramic electrochemical cells

Xiaoyu Wang[1,2], Zhaohui Cai[2], Zeping Chen[1,2], Donliang Liu[1,2], Wanqing Chen[1,2], Jianqiu Zhu[3], Wenhuai Li[1,2], Xixi Wang[4], Linjuan Zhang [3], Wei Wang[1,2], Chuan Zhou[1,2] ✉, Wei Zhou [1,2] ✉ & Zongping Shao [5] ✉

The key challenges for commercializing reversible proton ceramic electrochemical cells (R-PCECs) are the insufficient proton conductivity and inferior thermomechanical stability of oxygen electrodes in air with water vapor. We report a multielement micro-doped $BaCoO_{3-\delta}$-based perovskite material, in which disorder is induced in the ionic substructure to maximize the oxygen-water reaction activity. Atom probe tomography and density functional theory calculations reveal that reduced proton adsorption/diffusion energy barriers are triggered by homogeneous ion distributions in the perovskite oxide. Moreover, the thermally driven mild oxygen release can be further offset by beneficial proton uptake, thereby increasing the thermomechanical durability of the oxygen electrode. The resulting R-PCECs obtain a peak power density of 1.56 W cm$^{-2}$ and an electrolysis current density of 2.0 A cm$^{-2}$@1.3 V at 600 °C while demonstrating long-term stability exceeding 780 hours, with degradation rates of 19.3 and 16.9 µV h$^{-1}$ in fuel cell and electrolysis modes, respectively.

Solid oxide cells, offering high-efficiency power generation and green hydrogen production, represent a promising sustainable energy conversion technology[1,2]. Reversible proton ceramic electrochemical cells (R-PCECs) constitute a substantial advancement by reducing the operating temperature from 750–1000 °C to the intermediate range of 350–600 °C, thus mitigating material degradation and sealing challenges associated with high-temperature operation[3–9]. However, the commercialization of R-PCECs is hindered by oxygen electrodes lacking simultaneous high catalytic activity and reliable operational stability[10–12]. This challenge is particularly demanding given the requirements for H$^+$/O$^{2-}$/e$^-$ triple-conductivity while preserving the microstructural integrity in harsh humidified oxidizing environments[13,14].

The typical oxygen electrodes for R-PCECs predominantly have ABO$_3$-type cubic perovskite structures. The highly symmetric lattice generally engenders isotropic ion migration and displays a greater tendency for atomic orbital overlap, which leads to promising potential of proton conduction[15]. Nevertheless, investigations have revealed a marked discrepancy between theoretical predictions and practical applications, such as Ba$_{0.5}$Sr$_{0.5}$Co$_{0.8}$Fe$_{0.2}$O$_{3-\delta}$ (BSCF), which has a suboptimal protonic conductivity[16]. A plausible explanation for this discrepancy lies in the nanoscale heterogeneity of oxygen sublattice ordering, despite apparent macroscopic crystallographic regularity. Localized strain in B-site cations and the spatial distribution of redox-active transition metal species with different valences engender microscopic domains of various degrees of order/disorder. This nanoscale heterogeneity subsequently has profound implications for both the electrochemical kinetics and thermodynamic stability of materials.

[1]State Key Laboratory of Materials-Oriented Chemical Engineering, College of Chemical Engineering, Nanjing Tech University, Nanjing, China. [2]Suzhou National Laboratory, Suzhou, China. [3]Key Laboratory of Interfacial Physics and Technology, Shanghai Institute of Applied Physics, Chinese Academy of Sciences, Shanghai, China. [4]School of Environmental Science and Engineering, Nanjing Tech University, Nanjing, China. [5]Curtin Centre for Advanced Energy Materials and Technologies (CAEMT), Western Australian School of Mines (WASM), Curtin University, Perth, WA, Australia. ✉e-mail: zhouc@szlab.ac.cn; zhouwei1982@njtech.edu.cn; Zongping.Shao@curtin.edu.au

Previous investigations revealed that B-site cation substitution strategies could modulate both the electrocatalytic activity and structural resilience. Zhou et al.'s investigation indicated that the improved oxygen reduction reaction (ORR) activity of $SrCo_{0.8}Nb_{0.1}Ta_{0.1}O_{3-\delta}$ can be attributed to the enhanced disorder in its ionic arrangement, thus establishing correlations between local structural disorder and enhanced catalytic kinetics[17,18]. Conversely, the heterogeneous distribution of transition-metal elements induces pronounced, substantial thermal stress gradients during thermal cycling, resulting in electrode delamination. Consequently, optimization of the cationic distribution homogeneity represents a multifunctional approach for concurrently optimizing the catalytic activity and operational durability of oxygen electrodes materials[19].

Herein, we report a multielement micro-doped $BaCoO_{3-\delta}$-based perovskite engineered to induce disorder in its ionic substructure, maximizing the accessibility and utilization efficiency of catalytically active sites. Through atom probe tomography (APT) and first-principles density functional theory (DFT) calculations, $BaCo_{0.8}(Zr_{1/6}Ti_{1/6}Zn_{1/6}In_{1/6}Cu_{1/6}Mo_{1/6})_{0.2}O_{3-\delta}$ (BCZTZICM) exhibits dispersed elemental distribution, facilitating low proton diffusion energy barriers. Compositional homogenization at the atomic scale also results in diminished internal thermal stresses, resulting in a low thermal expansion coefficient (TEC) of $17.67 \times 10^{-6} K^{-1}$. Furthermore, X-ray absorption near-edge structure (XANES) and in-situ Fourier transform infrared (FT-IR) spectroscopy confirm an enhanced proton uptake capability, which synergistically mitigates the microstructural degradation induced by electrode cracking. A $Ni-BaZr_{0.1}Ce_{0.7}Y_{0.1}Yb_{0.1}O_{3-\delta}$ (BZCYYb)-supported single cell achieves a peak power density (PPD) of $1.56 W cm^{-2}$ and an electrolysis current density of $2.0 A cm^{-2}$@1.3 V at 600 °C, demonstrating the promising performance of this electrode design.

## Results

### Materials design and distribution characterization

Given the significant impact of B-site ionic composition on perovskite activity and stability, we categorized B-site cations into active elements (Fe, Co, Ni, etc.) and dopant elements (Zr, Y, Zn, Cu, etc.) to further refine the disorder characteristics within the entropy concept[20,21]. The mixed ionic-electronic conductor BSCF and the $H^+/O^{2-}/e^-$ triple conductor $BaCo_{0.4}Fe_{0.4}Zr_{0.1}Y_{0.1}O_{3-\delta}$ (BCFZY) were selected, both exhibiting high activity and serving as benchmarks. The B-site composition of BSCF comprises entirely active elements, whereas BCFZY represents a structure regulated by 10% Zr and 10% Y co-doping. Furtherly, the multi-element micro-doped materials with more diverse elemental regulation were designed, where B-site Co ions are modulated by 20% co-doping of (Zr, Ti, Zn, In, Cu, Mo). This configuration includes low valence of $Zn^{2+}$ and $Cu^{2+}$, isovalent $In^{3+}$, $Zr^{4+}$, and $Ti^{4+}$, and high valence $Mo^{6+}$, producing an overall valence effect consistent with 10% Zr and 10% Y co-doping. The design incorporates a gradient ionic radius distribution from 0.6 Å ($Cu^{2+}$) to 0.72 Å ($Zr^{4+}$), inducing controllable lattice strain without destabilization, while balancing covalency and oxygen vacancy formation through transition metal elements (Zn, Cu, Mo) and non-transition metal elements (Zr, Ti, In). This yielded the multi-element micro-doped BCZTZICM, which together with BSCF and BCFZY establishes a B-site elemental dispersion gradient design concept. Differs fundamentally from conventional high-entropy or single-dopant approaches. Instead of maximizing configurational entropy, multi-element micro-doping strategy distributes a low total doping concentration (20% of B-sites) across six distinct elements (Zr, Ti, Zn, In, Cu, Mo) to prioritize ionic dispersion. This design suppresses ion aggregation and disrupts defect clusters while preserving 80% active Co for high catalytic activity. The resulting homogeneous local environment, combined with moderate lattice strain from valence and ionic-radius diversity, enables uniform proton-transport pathways with low energy barriers and yields a low, near-linear thermal expansion. Ionic dispersion is the central mechanism, synergistically enhanced by valence diversity and controlled strain.

X-ray diffraction (XRD) Rietveld refinement reveals consistent *Pm-3m* space group symmetry across all the compositions, with Fig. 1a and Supplementary Fig. 1a-b demonstrating comparable lattice parameters: BSCF ($a = 3.986$ Å), BCFZY ($a = 4.116$ Å), and BCZTZICM ($a = 4.094$ Å). High-resolution transmission electron microscopy images of BCZTZICM (Fig. 1b) demonstrates lattice fringes of 0.288, 0.183, and 0.236 nm corresponding to the (011), (120), and (111) planes, respectively, which is consistent with the refinement results. Selected area electron diffraction (Supplementary Fig. 2) reveals the cubic superlattice structure of BCZTZICM along the [$\bar{1}$10] zone axis. Energy-dispersive spectroscopy (Supplementary Fig. 3) confirms the successful incorporation of dopants into the BCZTZICM lattice, with a homogeneous distribution.

Proton conduction bottlenecks in oxygen electrodes critically limit R-PCECs performance. State-of-the-art electrolytes ($BaZr_{0.8}Y_{0.2}O_{3-\delta}$, BZCYYb) exhibit proton conductivity of ~0.01–0.02 S cm⁻¹ at 600 °C under humidified conditions[5,22,23]. The proton conductivity of conventional oxygen electrode materials, as evidenced by measurements utilizing diverse techniques-such as the bulk diffusion coefficient testing[24] for (Ba/Sr)(Co/Fe/W)$O_{3-\delta}$@$PrBa_{0.5}Sr_{0.5}Co_{1.5}Fe_{0.5}O_{5+\delta}$ (~$10^{-5}$ S cm⁻¹ at 800 °C), hydrogen permeation[25] for $La_{0.8}Sr_{0.2}Sc_{0.5}Fe_{0.5}O_3$ ($1.1 \times 10^{-4}$ S cm⁻¹ at 600 °C), and the Hebb-Wagner DC polarization method[26] for $BaGd_{0.3}La_{0.7}Co_2O_{6-\delta}$ ($4.4 \times 10^{-5}$ S cm⁻¹ at 600 °C) -is consistently and significantly lower than that of state-of-the-art proton-conducting electrolytes. This severe conductivity mismatch confines electrochemical reactions to narrow triple-phase boundaries, restricting the effective reaction zone to approximately 1–2 μm near the electrode-electrolyte interface, limiting electrode utilization and overall cell performance[27]. Insufficient proton conductivity can be partially attributed to local ionic arrangement and valence inhomogeneity, as illustrated by the clustering and dispersion patterns of B-site ion arrangements in the R-PCECs structure shown in Fig. 1c-d. Proton conduction in perovskite oxide mechanisms exhibits temperature-dependent dominance. At elevated temperatures, protons can migrate via the vehicle mechanism as mobile OH⁻ species through oxygen vacancy sites. At intermediate temperatures characteristic of R-PCECs operation (450–600 °C), the prohibitively high activation energy (>1 eV) for OH⁻ transport renders this pathway unfavorable, and the Grotthuss mechanism predominates. Grotthuss mechanism occurs through thermally activated hopping between adjacent oxygen sites, with transport rates being critically dependent on local electronic environments and geometric constraints[7,28]. Compositional heterogeneity creates energy barriers limiting proton conductivity, while uniformly dispersed oxygen vacancies enable facile migration pathways (Fig. 1e-f).

FT-IR spectroscopy reveals symmetric B-O-B stretching (610 cm⁻¹) and asymmetric B-O-B' modes (625 cm⁻¹), showing increasing B-site cation asymmetric arrangement from BSCF through BCFZY to BCZTZICM (Supplementary Fig. 4)[29]. APT, which offers atomic-scale precision with high-spatial resolution compositional analysis[30]. Through APT characterization of dense needle-shaped samples prepared from the three materials, detailed ionic distribution information was obtained. Supplementary Fig. 5a-b reveals a modest increase in the oxygen content of BSCF, accompanied by pronounced Co and Fe depletion at a depth of 35 nm. BCFZY exhibits relatively large fluctuations in Co, Fe, and O concentrations, suggesting potential ionic clustering and heterogeneous valence distributions (Supplementary Fig. 6). Examination of the BCZTZICM ionic arrangement along the y- and z-axes (Fig. 1g and Supplementary Fig. 7) reveals a more homogeneous distribution without regional ionic enrichment. To gain deeper insights into the local chemical environment and potential nanoscale segregation tendencies, radial distribution function (RDF) analysis derived from APT was performed[30,31]. The key metric is the

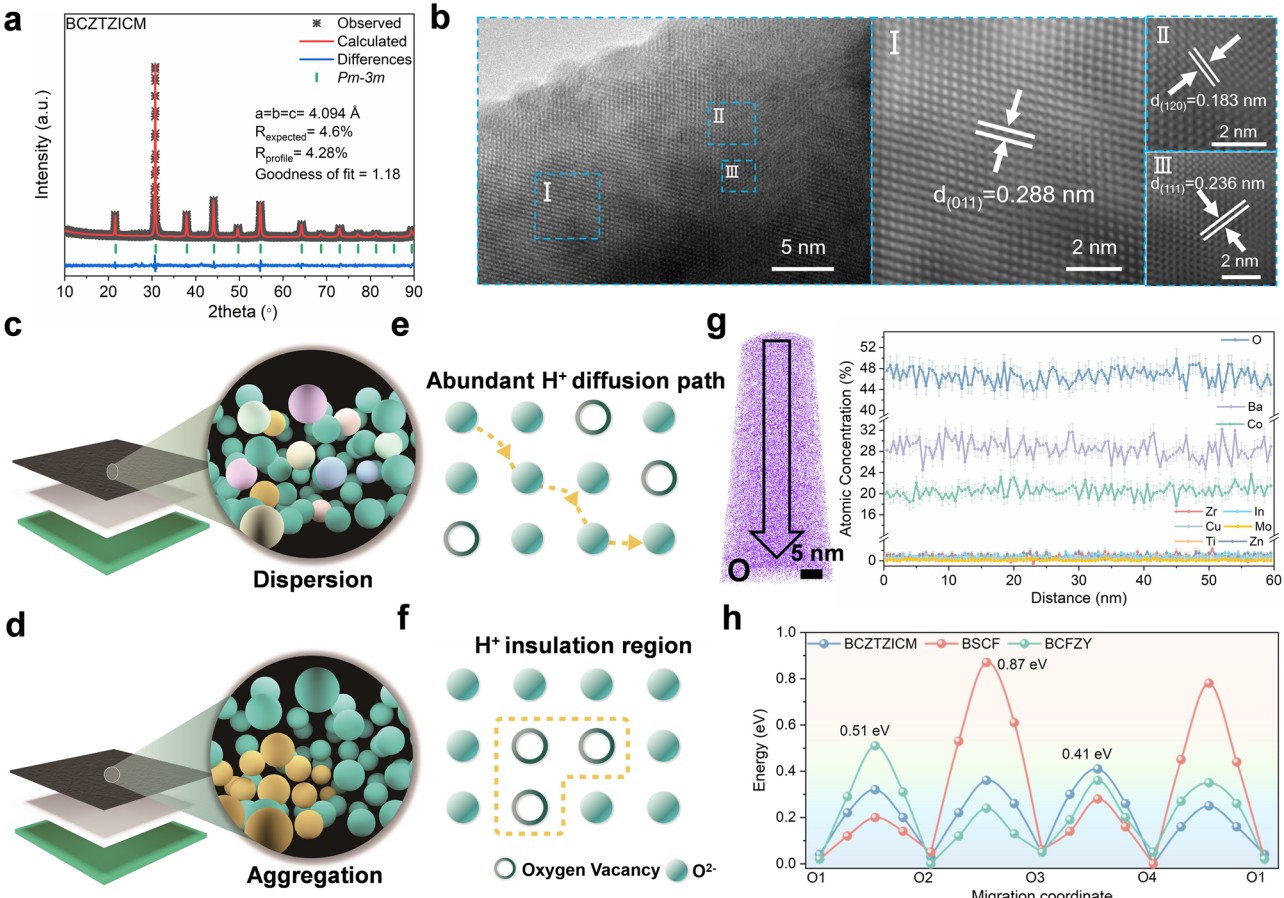

**Fig. 1 | Dispersed B-site doping eliminates proton diffusion bottlenecks in BCZTZICM perovskite electrodes. a** Refined XRD profiles of the prepared BCZTZICM. **b** HR-TEM image of BCZTZICM along (011), (120), and (111) plane. Structure diagram of R-PCECs, the amplification part is a brief sketch of the ion arrangement of (**c**) dispersed and (**d**) aggregated at B site of perovskite-type oxygen electrode. Schematic illustrating (**e**) abundant proton conduction pathways facilitated by discrete oxygen vacancies enabled through homogeneous ionic arrangement, and (**f**) regions of elevated proton diffusion barrier due to contiguous vacancies. **g** 3D reconstruction of the APT data showing mapping of the BCZTZICM sample with O distributions. Integrated line profiles show chemical composition alongside the z-axis of the analyzed needle. **h** Proton diffusion energies of BCZTZICM, BSCF, and BCFZY in the bulk. Source data for Fig. 1 are provided as a Source Data file.

proximity of the normalized concentration profiles to unity across various radial distances, which indicates similar association tendency between different ion species. As illustrated in Supplementary Fig. 8, with Co ions as the center, the normalized concentrations of high-abundance Co cations and oxygen anions approach unity, while trace-doped $In^{3+}$ and $Zr^{4+}$ ions display vibrant concentration profiles within the analyzed region (constrained by limited abundance), collectively indicating a remarkably homogeneous spatial distribution of Co ions, $O^{2-}$, and dopant ions. In contrast, BSCF suggests a slight tendency for Co ions to disperse from Fe, implying a preferential association among like ions (Co-Co and Fe-Fe) within the analyzed region, and $O^{2-}$ demonstrates a stronger association with Co than with Fe. For BCFZY, Co exhibits a preferential association with $Y^{3+}$, Fe ions and $O^{2-}$, while displaying a relative exclusion toward $Zr^{4+}$.

DFT calculations based on optimized structural models (optimized structures: Supplementary Fig. 9a-c, Supplementary Data 1) reveal that BCZTZICM exhibits the lowest proton diffusion barrier (Fig. 1h) along bulk transport pathways (Supplementary Fig. 10a-c). While BSCF contains low-barrier sites, high-barrier regions obstruct migration, limiting conductive pathways and the overall proton conductivity. Notably, BCZTZICM maintains highly-uniformity, with lower difference between maximum and minimum barrier sites, eliminating bottleneck effects and creating extensive networks of facile transport pathways that enable rapid proton diffusion in three dimensions.

Metal-oxygen bond length distributions analysis (Fig. 2a) reveals minor Co-O delocalization in BCZTZICM versus significant deviations in BSCF. Electron localization function (ELF) calculations (Fig. 2b-c, Supplementary Fig. 11) were used to assess the internal homogeneity. The standard deviation measurements confirm that the multi-doped systems exhibit increased uniformity of Co-O bonds and oxygen ions. Additionally, visualization diagrams along the (100) plane and bird's-eye view ELF maps are presented in Supplementary Fig. 12. These results demonstrate that the presence of dopant ions on the (100) plane, no significant polarization effects on oxygen ELF, indicating enhanced structural homogeneity in BCZTZICM and the mitigation of potential well effects from doping ions on carrier migration processes. In contrast, BSCF exhibits pronounced O-ELF polarization disparities (Supplementary Fig. 12b and e), resulting in the high ionic migration barrier sites.

BCZTZICM exhibits the highest average O-ELF value compared with BCFZY and BSCF, indicating enhanced oxygen locality that suppresses high-temperature oxygen desorption while preserving proton pathways (Fig. 2d). The oxygen vacancy formation energies demonstrate BCZTZICM's minimal energy variation between sites (Supplementary Fig. 9), confirming structural homogeneity (Fig. 2e). Thermogravimetric (TG) analysis demonstrates a mass loss of 1.12% for BCZTZICM and higher losses for BSCF and BCFZY, indicating an enhanced metal-oxygen bond energy of BCZTZICM. Based on room-

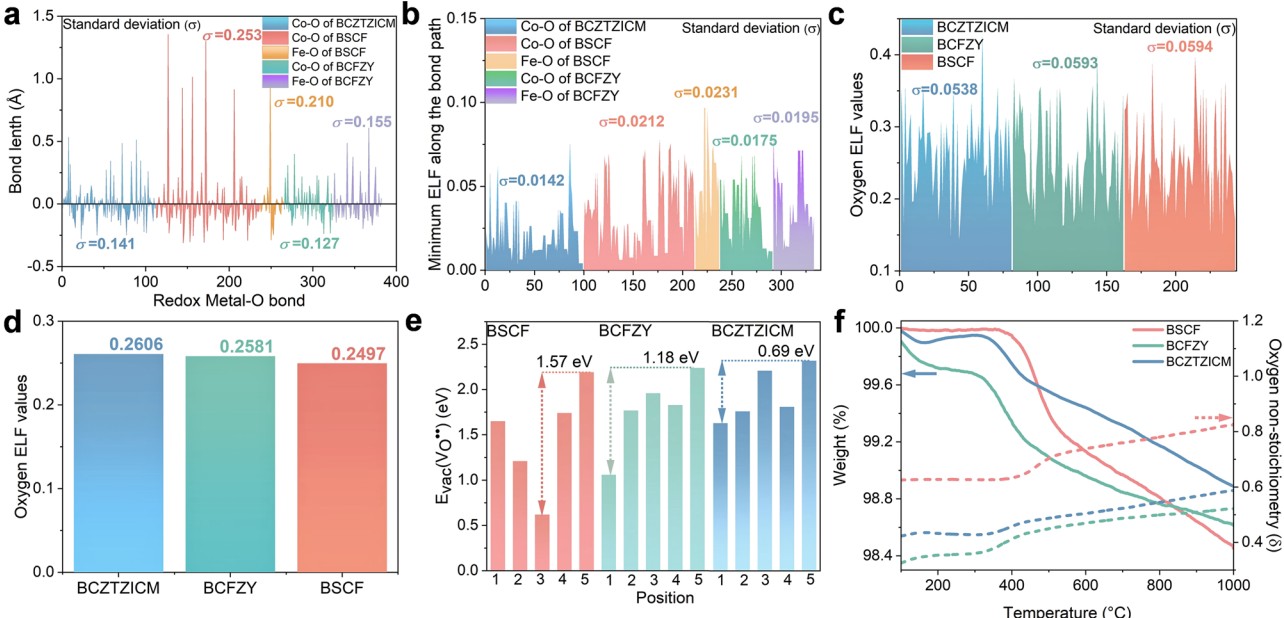

**Fig. 2 | Multielement micro-doping enhances structural homogenization.**
**a** Bond length distributions, (**b**) minima of the Electron localization function (ELF) along bond paths, and (**c**) O-ELF values for BCZTZICM, BSCF, and BCFZY, using standard deviation (σ) as the metric for internal uniformity. **d** Average O-ELF values, (**e**) oxygen vacancy formation energy, and (**f**) TG and oxygen non-stoichiometric ratio data of BCZTZICM BSCF and BCFZY samples. Source data for Fig. 2 are provided as a Source Data file.

temperature XANES, the oxygen non-stoichiometric ratios were obtained. BCZTZICM demonstrates a higher concentration of inherent oxygen vacancies compared to BCFZY, which is beneficial to surface activity (Fig. 2f)[28]. Oxygen temperature-programmed desorption ($O_2$-TPD) reveals minimal $O_2$ desorption peaks for BCZTZICM (Supplementary Fig. 13).

## Thermal-mechanical stability analysis

Thermal-mechanical compatibility represents a critical requirement for practical R-PCECs implementation, as thermal expansion mismatch between oxygen electrode ($20–25 \times 10^{-6} K^{-1}$) and electrolyte ($10–12 \times 10^{-6} K^{-1}$) components generates destructive stresses during operational temperature cycling[10]. Elevated Co/Fe content generates excessive intergranular thermal stress, resulting in electrode fragmentation. Integration of APT results with theoretical calculations reveals that materials exhibiting severe ionic localization (e.g., BSCF) readily form excessive thermal stress ultimately inducing electrode cracking (Fig. 3a).

Thermal expansion tests in Fig. 3b presents TECs in dry air (300–800 °C): BSCF ($26.08 \times 10^{-6} K^{-1}$), BCFZY ($22.56 \times 10^{-6} K^{-1}$), and BZCYYb ($11.98 \times 10^{-6} K^{-1}$). BCZTZICM achieves a remarkably low TEC of $17.67 \times 10^{-6} K^{-1}$ (300–800 °C), lower than those of most reported oxygen electrodes (Supplementary Table 1). BCZTZICM exhibits near-linearity thermal expansion behavior, contrasting with BSCF and BCFZY deviations from linearity above 400 °C. This deviation results from thermally induced lattice oxygen loss, reducing cation coordination numbers and increasing cation electrostatic repulsion, leading to anomalous expansion behavior and potential mechanical instability[19]. Multielement B-site doping in BCZTZICM effectively suppresses lattice oxygen loss through uniform local ion coordination environment, yielding near-linear behavior.

XANES provides quantitative analysis of valence changes during thermal treatment under different atmospheric conditions. Co K-edge and Fe K-edge XANES spectra for BCZTZICM, BCFZY, and BSCF reveal temperature- and atmosphere-dependent valence transitions (Fig. 3c and Supplementary Fig. 14a-b and 15b-c). Calibration using CoO, $Co_3O_4$, and $LaCoO_3$ for Co and $Fe_4Nb_2O_9$ and $SrFeO_3$ for Fe

(Supplementary Fig. 15a) established precise relationships between the absorption edge positions and average oxidation states. As shown in Supplementary Figs. 16a-c and 17a-b, the changes in the lattice oxygen content can be accurately determined.

Under dry air conditions, BCZTZICM demonstrates minimal lattice oxygen content change (−0.075) compared with substantial losses for BSCF (−0.247) during heating to 600 °C (Fig. 3d), indicating lower thermal expansion, which were calculated using the equation in Supplementary Note 1. Notably, steam-treated samples exhibit elevated B-site cation valences. This oxidative response upon proton uptake originates from a distinct mechanism: the lowering of antibonding O $2p$ states hybridized with transition metal $3d$ orbitals, coupled with modifications to the local coordination environment induced by protonic defect incorporation (Supplementary Note 2)[32]. This mechanism has been experimentally validated through *operando* X-ray absorption spectroscopy (XAS) studies on analogous perovskite systems, depending on the specific electronic structure characteristics (Supplementary Note 2)[33,34]. After normalization for elemental composition, the lattice oxygen content variations derived from XANES analysis quantify the proton uptake capacity (Fig. 3d). BCZTZICM exhibits a competitive proton uptake capability under humid conditions, with a lattice oxygen content increase of 0.056, compared with only 0.042 for BSCF, when switching from dry to humidified environments at 600 °C. The multi-element micro-doping, by maximizing ionic dispersion, creates this enhanced structural homogeneity, which in turn leads to a greater availability of uniform active sites, resulting in significantly improved proton uptake capability.

Rigorous thermal cycling tests assessed BCZTZICM electrodes durability under conditions simulating realistic R-PCECs operation. Testing protocols involved 40 rapid thermal cycles between 400 °C and 600 °C with a heating rate of 20 °C min$^{-1}$ and a cooling rate of 10 °C min$^{-1}$, representative of aggressive startup/shutdown scenarios in grid-responsive applications[35]. Chemical compatibility assessment between BCZTZICM and BZCYYb (Supplementary Fig. 18) reveals no emergent peaks or shifts following 1000 °C co-sintering, confirming compatibility. High-temperature XRD (Supplementary Fig. 19) and patterns obtained 200 hours post-calcination at 600 °C

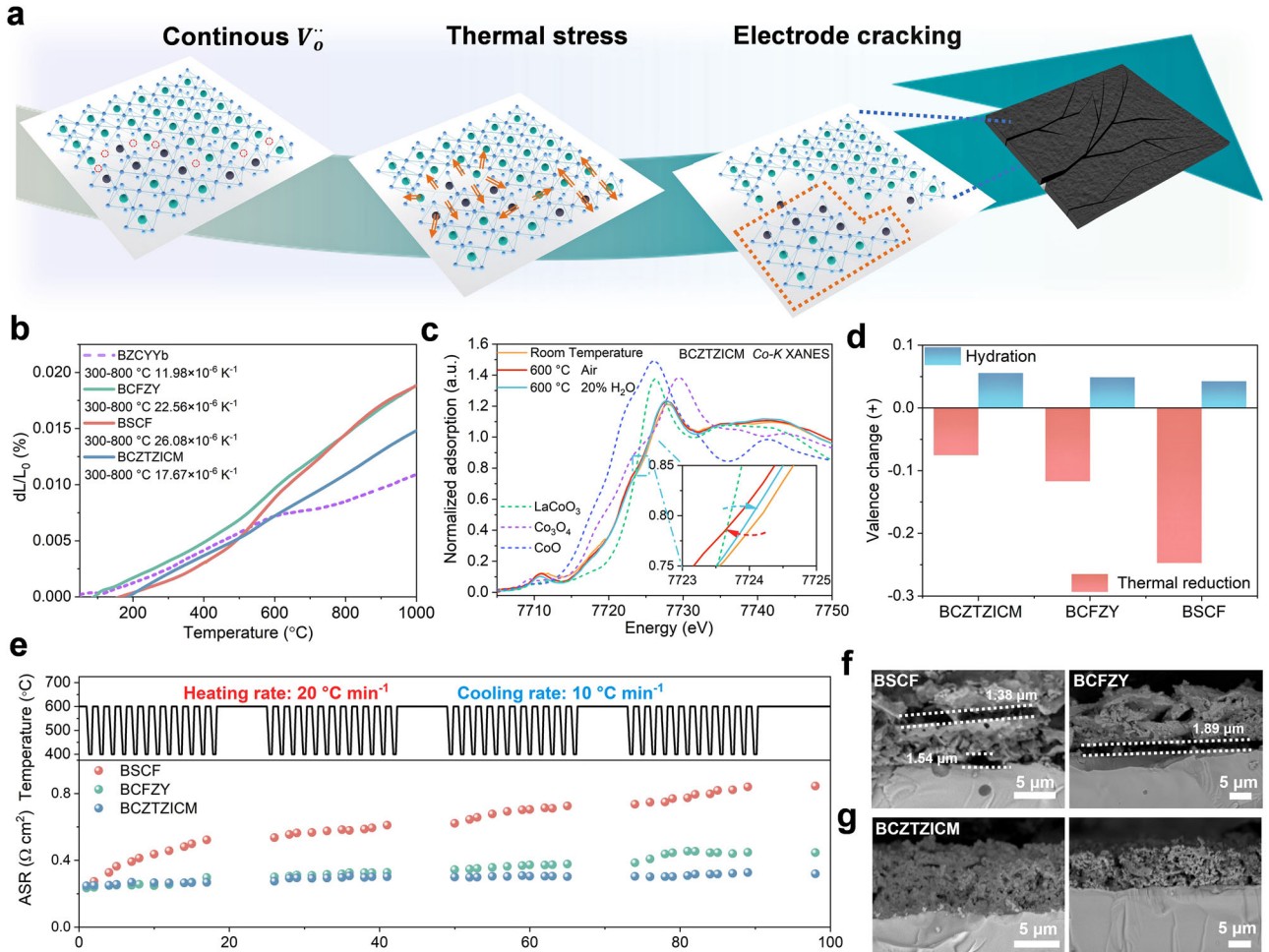

**Fig. 3 | Suppressed thermal expansion and improved cycling stability through compositional homogenization. a** Schematic of electrode tearing driven by increased internal thermal stress resulting from continuous oxygen vacancies. **b** Thermal expansion coefficient curves measured in air over 300–800 °C. **c** Co K-edge XANES spectra of BCZTZICM samples: as-prepared (25 °C), dried at 600 °C, and treated under 20% $H_2O$ atmosphere. **d** Average B-site valence changes derived from XANES analysis. **e** The Rp response of BCZTZICM, BCFZY, and BSCF-based symmetric cell electrodes during 40 thermal cycles between 600 °C and 400 °C. Cross-sectional image of cells after cycling (**f**) BSCF, BCFZY, and (**g**) BCZTZICM-based electrode symmetric cell with cracks inside the electrode bulk and at the electrode-electrolyte interface. Source data for Fig. 3 are provided as a Source Data file.

(Supplementary Fig. 20) demonstrate structural robustness. The steam tolerance of BCZTZICM was evaluated by high-temperature XRD during cyclic transitions between dry and humid air atmospheres. As shown in Supplementary Fig. 21, the material exhibits only reversible lattice expansion/contraction, without emergence of secondary phases.

Area specific resistance (ASR) increase over 100 hours and 40 complete cycles (Fig. 3e) demonstrated competitive electrode stability: BSCF showed polarization resistance (Rp) 256% degradation (6.4% per cycle), BCFZY 85% (2.125% per cycle), while BCZTZICM exhibited only 28% (0.58% per cycle). The initial, 20 and 40 cycle polarization impedances are shown in the Supplementary Fig. 22. This substantially lower degradation rate confirms the promising long term and multi-thermal cycle stability of the BCZTZICM, enabling reliable operation under demanding conditions. These results demonstrate competitive mechanical durability enabling reliable operation under demanding conditions. Post-cycling scanning electron microscopy (SEM) cross-sectional analysis reveals structural origins of the enhanced durability. BSCF and BCFZY electrodes (Fig. 3f) exhibit extensive cracking networks with 1–2 μm crack widths, complete delamination from electrolyte interfaces, and significant porosity changes, indicating mechanical failure. Surface laminar cracks and internal fissures

correlate with elevated thermal stress. In contrast, Fig. 3g shows that BCZTZICM maintains structural integrity with minimal visible defects, confirming its competitive thermomechanical compatibility. The thermal cycling conditions and the corresponding crack widths observed in post-test microstructural analysis for reference electrodes are summarized in Supplementary Table 2. Enhanced thermo-mechanical stability originates from two synergistic mechanisms: the homogeneous ionic distribution enables uniform oxygen release, yielding a reduced and near-linear thermal expansion coefficient; concurrently, proton uptake capability suppresses Co reduction while the accompanying chemical expansion compensates for thermal contraction during cooling.

**Proton uptake conduction dynamics**
Proton transport in perovskite-based oxygen electrodes follows the Grotthuss mechanism in both operational modes. Surface proton transport, in contrast, originates from the proton uptake process through $H_2O$ dissociative chemisorption to form protonic defects. Fundamental distinctions arise in proton sources between operational modes: in fuel cell mode, protons originate from hydrogen dissociation at the hydrogen electrode, transporting through electrolyte, and forming water at the oxygen electrodes, while water vapor generated

as reaction product undergoes subsequent proton uptake, augmenting electrode protonic conductivity. Electrolysis mode generates protons exclusively through water dissociation at the oxygen electrode, with subsequent transport to the hydrogen electrode. The overall efficiency critically depends on electrode ability to rapidly adsorb water vapor, facilitate dissociative chemisorption, and transport protons into the bulk. Physical absorption assessment through TPD-mass spectrometry following water vapor exposure (250 °C, 20% $H_2O$, 2 hours). BCZTZICM exhibits enhanced surface physisorption and water storage capacity, shown in Fig. 4a. DFT calculations provide molecular-level insights into the enhanced water interaction mechanisms. BCZTZICM surfaces exhibit favorable adsorption energetics compared with those of BSCF and BCFZY, with lower hydration reaction energetics, as shown in Fig. 4b.

In-situ FT-IR spectroscopy during high-temperature water exposure provides real-time hydration kinetics and intermediate species formation monitoring. Hydroxyl formation was monitored via O-H stretching vibrations (3200–3700 cm$^{-1}$) following 10% water vapor introduction at 600 °C[21]. BCZTZICM demonstrates rapid chemisorption kinetics, achieving steady-state hydroxyl formation within 5 minutes, like BCFZY, versus more than 45 minutes for BSCF. Upon dry air purging, the hydroxyl elimination kinetics reveal proton mobility.

BCZTZICM shows the rapid disappearance of the hydroxyl peak (complete elimination within 15 minutes), indicating efficient transport of protons from bulk positions to surface sites, contributing to the ionic conductivity. BSCF retains significant hydroxyl signatures after 45 minutes, suggesting severely limited proton mobility.

## Electrochemical performance evaluation

Electronic conductivity measurements (Supplementary Fig. 23) reveal the characteristic of BSCF non-monotonic temperature dependence, directly correlated with severe oxygen loss. Excessive oxygen vacancies annihilate electron holes, reducing carrier concentration and conductivity at high temperatures. In contrast, BCZTZICM maintains stable oxygen content at operating temperatures, exhibiting near-linear temperature-dependent conductivity dominated by thermally activated electron hopping. This demonstrates that lattice oxygen desorption in BCZTZICM exhibits a stochastic desorption characteristic. In contrast, the non-linear conductivity responses about temperature in BSCF and BCFZY are associated with the inhomogeneity of local structures.

Electrical conductivity relaxation (ECR) measurements were conducted to extract oxygen surface exchange coefficients ($k_{chem}$) and bulk diffusion coefficients ($D_{chem}$) (Supplementary Fig. 24). Notably,

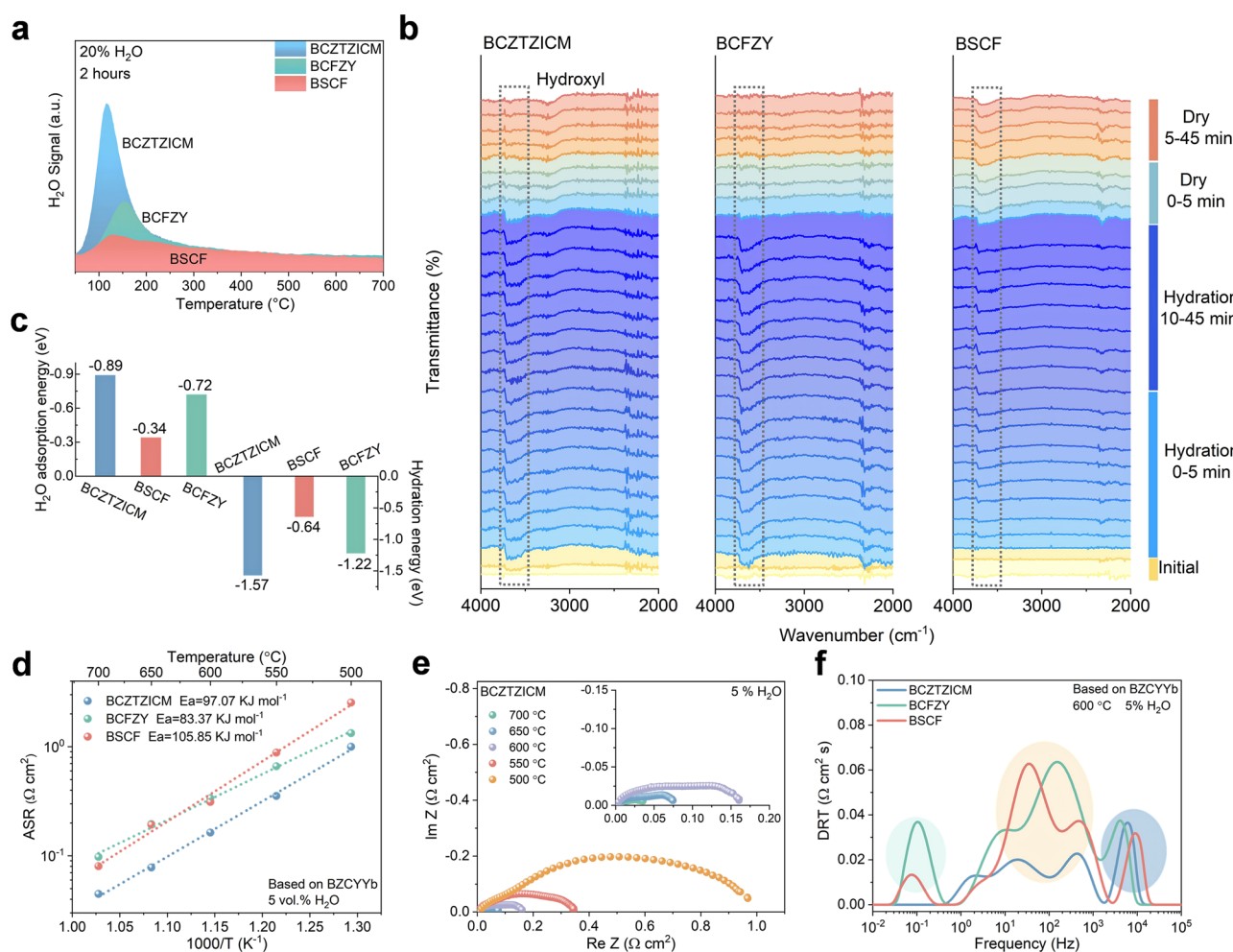

**Fig. 4 | Water adsorption kinetics and electrochemical performance of compositionally homogenized perovskite electrodes. a** $H_2O$-TPD plots of BCZTZICM, BCFZY, and BSCF between 25 and 1000 °C. **b** FT-IR spectra of BCZTZICM, BCFZY, and BSCF at 600 °C adsorption and desorption process. **c** Theoretically calculated water vapor adsorption energy and hydration reaction energy of BCZTZICM, BCFZY, and BSCF. **d** Arrhenius plots of BCZTZICM, BCFZY, and BSCF between 700 and 500 °C based on BZCYYb in 5% $H_2O$-air and corresponding (**e**) EIS curve of BZCYYb based-supported symmetrical cell with BCZTZICM electrodes in 5% $H_2O$-air at 500–700 °C. **f** DRT plots of BCZTZICM, BCFZY, and BSCF electrodes in 5% $H_2O$-air at 600 °C. Source data for Fig. 4 are provided as a Source Data file.

BCZTZICM shows inferior oxygen activity versus BSCF under dry air. Electrochemical impedance spectroscopy (EIS) of symmetric cells determines of electrode-specific contributions. Dry-air testing of $Gd_{0.2}Ce_{0.8}O_{1.9}$ (GDC) based cells eliminated the confounding effects of proton conduction. An Arrhenius plot (Supplementary Fig. 25a) reveals unfavorable BCZTZICM ORR kinetics under dry conditions due to enhanced metal-oxygen bonding from multi-doping, consistent with DFT calculations. BSCF and BCFZY exhibit better intrinsic ORR activity than BCZTZICM (Supplementary Fig. 25b-d). However, in controlled humidified environments (5% $H_2O$ in air), BCZTZICM achieves remarkably low ASR of 0.045, 0.078, 0.163, 0.354, and 0.998 $\Omega\,cm^2$ across the complete operational temperature range of 700–500 °C (Fig. 4d-e), substantially outperforming BCFZY (0.337 $\Omega\,cm^2$) and BSCF (0.311 $\Omega\,cm^2$) at 600 °C based on the BZCYYb electrolyte (Supplementary Fig. 26a-b). The enhanced proton transport kinetics become pronounced at lower temperatures (500 °C), where proton conduction dominates. Disparities between GDC and BZCYYb symmetric cells testing demonstrate BCZTZICM's advantages in protonic conduction. In addition, steam introduction reduces the ASR versus dry air (Supplementary Fig. 27), through accelerated surface exchange and bulk conduction.

Distribution of relaxation times (DRT) analysis deconvolves impedance spectra into distinct processes contributions. High-frequency regions (>1000 Hz) reflect charge transfer processes at electrode-electrolyte interfaces, intermediate-frequencies (1–1000 Hz) correspond to bulk ion transport within electrode, and low-frequencies (<1 Hz) indicate gas-phase diffusion limitations[36,37]. As shown in Fig. 4f, BCZTZICM exhibits substantially reduced contributions across all frequency ranges, with dramatic improvements in the mid-frequency regions corresponding to bulk proton transport. The

weaker low-frequency response reflects enhanced surface proton uptake, aligning with the in-situ FT-IR and $H_2O$-TPD results. This confirms atomic-scale compositional homogenization creates extensive facile proton transport pathways optimizes proton uptake, and retains sufficient intrinsic oxygen activity. Comparison with state-of-the-art electrode materials confirm the promising performance of BCZTZICM (Supplementary Fig. 28).

Single cells with Ni-BZCYYb hydrogen electrodes, BZCYYb electrolytes, and BCZTZICM oxygen electrodes underwent I-V and EIS testing. BCZTZICM oxygen electrodes achieve a 1.3 W cm$^{-2}$ PPD at 650 °C with a 0.07 $\Omega\,cm^2$ polarization resistance (Supplementary Fig. 29) in fuel cell mode, versus BCFZY (0.9 W cm$^{-2}$) and BSCF (0.81 W cm$^{-2}$), as shown in Supplementary Figs. 30–31. The current density in electrolysis mode is 1.8 A cm$^{-2}$ at 650 °C (Supplementary Fig. 32a), versus BSCF (1.32 A cm$^{-2}$) and BCFZY (1.29 A cm$^{-2}$) (Supplementary Figs. 33–34), respectively. Supplementary Fig. 35 presents cross-sectional SEM images of the comparable electrolyte (13–14 µm) and electrode thickness (10–11 µm).

R-PCECs incorporating thin film electrolytes were fabricated following the methodology described by Duan et al.[4,38]. The measurements in Fig. 5a yield PPDs of 1.56, 1.24, 0.93, and 0.64 W cm$^{-2}$ at 600, 550, 500, and 450 °C, respectively. As shown in Fig. 5c, the total cell polarization resistance reaches only 0.11 $\Omega\,cm^2$ at 600 °C. Electrolysis measurements (Fig. 5b) at a thermoneutral voltage (1.3 V) yield results of 2.0–0.6 A cm$^{-2}$ at 600–450 °C. The enhanced oxygen evolution reaction (OER) activity reflects the enhanced proton uptake capabilities and the optimized proton transport properties.

Faradaic efficiency measurements confirm hydrogen production selectivity with minimal parasitic losses. The efficiencies reach 85.4 ± 0.55% at optimal operating conditions (1.2 A cm$^{-2}$, 550 °C) and

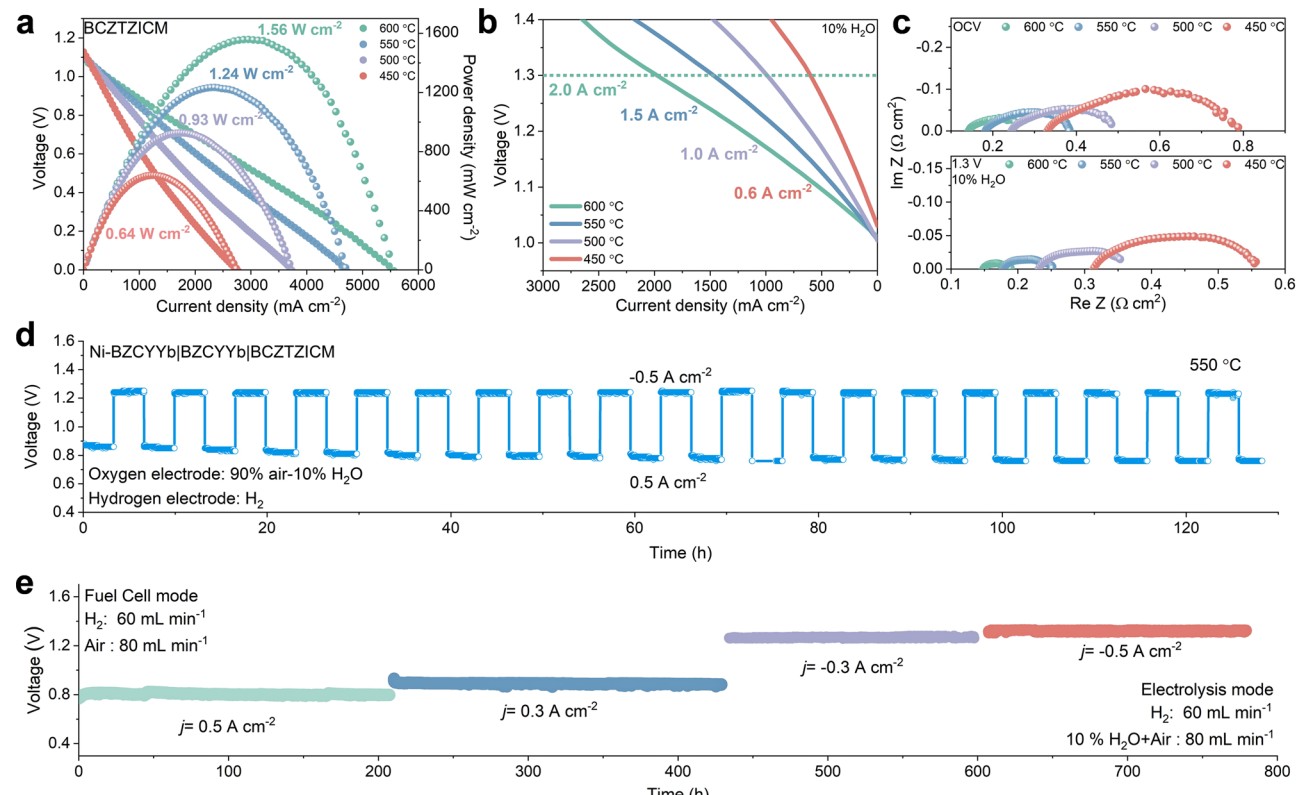

**Fig. 5 | Reversible fuel cell and electrolysis performance with long-term operational stability.** I-V curves of the cells with the configuration of Ni-BZCYYb | BZCYYb | BCZTZICM in (**a**) fuel cell mode under $H_2$/Air and (**b**) electrolysis mode under $H_2$/10% $H_2O$-Air operation at 500–650 °C, and its (**c**) corresponding impedance with active area of 0.45 cm$^{-2}$. **d** Reversible operation stability over 120 h. **e** Long-term operational stability of the button cell with BCZTZICM oxygen electrodes operation at 550 °C in fuel cell and electrolysis mode. All voltage values reported in this work were not iR corrected. Source data for Fig. 5 are provided as a Source Data file.

remain above 83% even at aggressive current densities (1.6 A cm$^{-2}$). The corresponding hydrogen production rate of 9.264 mL cm$^{-2}$ min$^{-1}$ and electrical energy-to-chemical energy conversion efficiency reached 58-68%, demonstrating capability for high-purity hydrogen production applications (Supplementary Fig. 36)[39]. Supplementary Fig. 37 presents cross-sectional SEM images of the tested cell featuring electrolyte (~7 μm) and electrodes (9–10 μm). Performance and impedance comparison establishes BCZTZICM among higher-performing R-PCECs systems (Supplementary Fig. 38 and Supplementary Table 3). A comparison of the low-temperature (450 °C) performances is shown in Supplementary Table 4. Temperature-dependent performance analysis reveals competitive kinetics across the complete operational window, with particularly competitive low-temperature capabilities that enable operation below 500 °C, providing advantages for thermal management requirements in distributed energy conversion. Figure 5d demonstrates reversible operation between fuel cell mode (0.5 A cm$^{-2}$) and electrolysis mode (−0.5 A cm$^{-2}$) over 120 h, showing competitive flexibility with minor hysteresis, confirming suitability for grid-responsive energy storage[40]. The long-term galvanostatic testing at 550 °C (Fig. 5e) demonstrates stable performance over 780 h, with degradation rates of 19.3 and 16.9 μV h$^{-1}$ in fuel cell mode and electrolysis mode, respectively. The degradation originates from seal failure induced by silver sealant oxidation, creep, thermal stress mismatch, and mechanical deterioration under high-temperature and humid atmospheres. Recent studies on oxygen electrode materials and corresponding long-term stability in R-PCECs summarized in Supplementary Table 5. The reliable stability observed at reduced temperatures, coupled with the competitive activity of BCZTZICM, highlights the promising application prospects of R-PCECs for efficient and sustainable energy conversion.

## Discussion

In conclusion, we developed a multielement micro-doping strategy for BCZTZICM oxygen electrodes with competitive ORR/OER activities and stability. High ion dispersion achieves a low TEC of $17.67 \times 10^{-6}$ K$^{-1}$ and highly hydration capacity, facilitating oxygen and proton mobility. BCZTZICM exhibits reliable thermal-mechanical stability in a 40-cycle thermal cycle test without noticeable degradation. Single cells achieve a PPD of 1.56 W cm$^{-2}$ and an electrolytic current density of 2.0 A cm$^{-2}$. This work establishes atomic-scale compositional homogenization through multielement doping as a strategy for optimizing multiple properties of ceramic electrodes and provides a paradigm for developing advanced materials. The combination of high power density, efficient electrolysis operation, promising durability, and reduced temperature requirements position R-PCECs systems as leading candidates for next-generation energy storage and conversion technologies.

## Methods
### Materials synthesis

The mentioned materials BCFZY, BSCF, and BCZTZICM were synthesized using sol-gel methods, consistent with literature description[41]. In the case of BCZTZICM, the ionic sources employed for dissolution and thorough mixing in the solution include Ba(NO$_3$)$_2$-99%, Co(NO$_3$)$_2$·6H$_2$O-98.5%, Zn(NO$_3$)$_2$·6H$_2$O-99%, Cu(NO$_3$)$_2$·3H$_2$O-99% (above purchased from Sinopharm Chemical Reagent Co., Ltd., analytical grade), Zr(NO$_3$)$_4$·5H$_2$O-99.99% (purchased from Shanghai Macklin Biochemical Co., Ltd, analytical grade), C$_{16}$H$_{36}$O$_4$Ti-99%, In(NO$_3$)$_3$·2H$_2$O-99.9% (purchased from Shanghai Aladdin Biochemical Co., Ltd., analytical grade), and H$_8$MoN$_2$O$_4$-99.99% (purchased from Alfa Aesar (China) Chemical Co., Ltd.). Ethylenediaminetetraacetic acid (EDTA) and citric acid monohydrate (CA) were used as complexing agents, and ammonia hydroxide adjusted the pH to about 7. The overall molar ratio of CA: EDTA: total metal ions was maintained at 2:1:1. Sr(NO$_3$)$_2$-99%, Y(NO$_3$)$_3$·6H$_2$O-99%, and Fe(NO$_3$)$_3$·9H$_2$O-98.5% used for synthesizing BSCF and BCFZY were purchased from Sinopharm

Chemical Reagent Co., Ltd. The precursor BZCYYb was obtained via solid-state reaction method using the oxide as a cation source and ethanol as a solvent. BaCO$_3$-99%, ZrO$_2$-99%, CeO$_2$-99.9% were purchased from Sinopharm Chemical Reagent Co., Ltd., and Y$_2$O$_3$-99.99%, Yb$_2$O$_3$-99.99% were purchased from purchased from Shanghai Aladdin Biochemical Co., Ltd. The precursors were heated at 180 °C for 5 h, then calcined at 1000 °C for 5 h in a muffle furnace and annealed to obtain the desired powder. GDC powder was supplied by NexTech Materials, Ltd.

### Cell fabrication

Hydrogen electrode supported single cells were prepared by co-pressing NiO+BZCYYb (60:40 wt.% ratio) with BZCYYb (0.012 g) at 3 Mpa and calcining at 1475 °C for 10 h. The perovskite powder was mixed uniformly with isopropyl alcohol-99.7%, ethylene glycol-99.5%, and propylene glycol solvents-99% (purchased from Sinopharm Chemical Reagent Co., Ltd., analytical grade) using ball milling at 400 rpm for 30 min. Then the resulting mixture was sprayed on the electrolyte surface and co-sintered at 1000 °C for two h to obtain full cells with 0.45 cm$^{-2}$ active area. The symmetrical cells were pressed at 3 Mpa from BZCYYb powder mixed with 1 wt.% NiO for sinterability and calcined at 1475 °C for 10 h, with active area of 1.094 cm$^{-2}$.

### Characterization

Room-temperature X-ray diffraction (XRD) data were obtained by the Bruker D8 Advance, and its high-temperature component (Rigaku D/max 2500 V) was used to test the crystal structure of the sample at high temperatures. The material's microstructure, representative lattice diffraction fringes, and elemental distributions were obtained through high-resolution transmission electron microscopy (HR-TEM, JOEL, JEM-2100) with energy-dispersive spectroscopy (EDS). Specimen tips were fabricated via focused ion beam (FIB, Thermo Fisher Scientific, Helios CX) milling, followed by atom probe tomography (APT, LEAP 6000 XR) analysis under vacuum chamber employing laser-assisted field evaporation to acquire elemental composition and three-dimensional atomic positions. Scanning electron microscopy (SEM, ZEISS, Sigma 560, ZEEPTOOLS, ZEM15C) was employed for post-analysis characterization of the cells cross-sectional morphology. The thermal expansion coefficient (TEC) of samples from 25 °C to 1000 °C was obtained by a 402 °C dilatometer, NETZSCH. For Fourier Transform Infrared (FT-IR) spectral measurements, powders were mixed with KBr and then analyzed on a Spectrum 3 spectrometer (PerkinElmer) with high-temperature assemblies in the frequency range of 500–4000 cm$^{-1}$ with a resolution of 2 cm$^{-1}$. Hard X-ray absorption spectroscopy was performed at the Beamline station BL11B of the Shanghai Synchrotron Radiation Facility. The acquired hard XAS data were processed according to the standard procedures using the ATHENA module implemented in the IFEFFIT software packages[42]. The photon energy of the Co K-edge and Fe K-edge was calibrated using metal foils as references. A thermogravimetric (TG, NETZSCH, STA 449 F3) analysis was employed to investigate the weight loss of the materials at the range of 100 and 1000 °C. H$_2$O-temperature programmed desorption (H$_2$O-TPD) was conducted to observe the adsorbed H$_2$O desorption temperature and desorption amount of BCZTZICM, BSCF, and BCFZY through mass spectrum (Hiden Analytica, HPR20). In-situ near-ambient pressure X-ray photoelectron spectroscopy (NAP-XPS, SPECS) was performed in 0.3 mbar air atmosphere. During the in-situ process from 25 °C to 600 °C, continuous data acquisition was performed. The final scan collected after the sample was cooled to 25 °C and fully stabilized was designated as the room-temperature data. Figures 1c-d and 3a were created using PowerPoint.

### Computational methods

The density functional theory (DFT) calculations were conducted using the Vienna Ab initio Simulation Package (VASP 5.4.4) with two

Hygon 7490 64-core processors[43,44]. The electronic exchange and correlation were treated using the Perdew-Burke-Ernzerhof (PBE) functional within the generalized gradient approximation (GGA) framework[45]. Core-valence electron interactions were described by the projector-augmented wave (PAW) pseudopotentials[46,47], and the electronic wavefunctions were expanded in a plane wave basis set with a kinetic energy cutoff of 500 eV. The perovskite structures were constructed with a $3 \times 3 \times 3$ supercell containing 27 $ABO_3$ formula units, and lattice parameters of 11.8720 Å×11.8720 Å×25.8934 Å ($\alpha = \beta = \gamma = 90°$). All model coordinates related to calculation are shown in Supplementary Data 1. Electronic convergence was achieved when the energy difference between successive iterations was less than $10^{-5}$ eV, while geometric optimization was considered complete when the forces on individual atoms were below 0.05 eV/Å. Brillouin zone sampling was performed using the Monkhorst-Pack scheme[48] with a k-point mesh of $3 \times 3 \times 1$. Spin polarization effects were incorporated in all calculations. Atomic coordinates of BCZTZICM, BSCF, and BCFZY for optimization calculation are shown in Supplementary Data 1.

## Electrochemical measurements

Electrochemical Impedance Spectroscopy (EIS) of the symmetrical cells and single cells was tested by potentiostat (Solartron 1287) and a galvanostat combined with frequency response analyzer (Solartron 1260 A) from 1 MHz to 0.01 Hz under an open circuit voltage, and the impedance of the electrolysis cell was measured at a nominal voltage of 1.3 V. The I-V and I-P curves were collected by source meter (Keithley 2440). During the test, dry $H_2$ was fed into the fuel side at a flow rate of 80 ml min$^{-1}$ (STP), while the oxygen electrodes were fed with flowing air at a flow rate of 80 ml min$^{-1}$ (STP). The Faradaic efficiency for electrolysis was quantified using online gas chromatography (GC, PANNA, A60 pro). During the measurement, the oxygen electrode side was supplied with 60 mL min$^{-1}$ flowing air (20% $H_2O$), while the hydrogen electrode side was purged with a flowing argon stream, which acted as a carrier gas directed to the GC for quantitative analysis of the produced hydrogen. The cell was stabilized for 20 minutes at each current density (0.8, 1.2, and 1.6 A cm$^{-2}$) before data collection. The GC chromatograms showed a distinct hydrogen peak, without observable nitrogen or oxygen peaks, confirming the reliable sealing of the cell and the absence of gas cross-leakage. To ensure statistical reliability, the measurement at each current density was repeated five times. The FE and energy efficiency was calculated referring following formula:

$$FE(\%) = \frac{n_{H_2, measured}}{n_{H_2, theoretical}} = \frac{n_{H_2, measured} \times (n \times F)}{I} \times 100\%$$

$$EEE(\%) = \frac{\Delta H_{H_2, LHV} \times n_{H_2, measured}}{I \times V} = \frac{n_{H_2, measured} \times (n \times F)}{I \times V} \times 100\%$$

Where $n_{H_2, measured}$ is the measured hydrogen production rate (mol s$^{-1}$), $n$ is 2, $F$ is the Faradaic constant (96,485 C mol$^{-1}$) and I is the applied current (A), $\Delta H_{H_2, LHV}$ is the lower heating value reaction enthalpy for steam electrolysis, with a value of 241.8 kJ mol$^{-1}$, and V is the corresponding voltage (V)[4,13,39].

## Data availability

The datasets analyzed and generated during the current study are included in the Article and Supplementary Information. Source data are provided with this paper.

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

## Acknowledgements

This work was financially supported by the National Natural Science Foundation of China (No. 22278203 (W.Z.), No. 22279057 (W.W.)) and Basic Research Program of Jiangsu (BK20250535 (C.Z.)). The authors acknowledge the technical and scientific support of the atom probe tomography and project financial support provided by Suzhou Laboratory. The authors are grateful for the technical support for Dr. Yifan Li of Nano-X from Suzhou Institute of Nano-Tech and Nano-Bionics, Chinese Academy of Sciences (SINANO).

## Author contributions

W.Z. and C.Z. supervised the project and conceived the idea. Z.H.C. supervised the theoretical calculation study. X.Y.W. designed and synthesized the materials and conducted the electrochemical and structural characterizations. J.Q.Z. and L.J.Z. performed XANES test. W.H.L. and X.Y.W. analyzed XANES data. D.L.L. performed the XRD Rietveld refinement. X.Y.W., C.Z., W.H.L., Z.P.C., W.Q.C., W.W., W.Z., Z.P.S. discussed the experimental results. X.Y.W., C.Z., W.W., W.Z., Z.P.S. wrote and edited the manuscript with input from all authors. All authors discussed the results and assisted during manuscript preparation.

## Competing interests

The authors declare no competing interests.
