## [Transparent Peer Review file · Nature Communications]

Highly ionic-dispersed oxygen electrode for reversible proton ceramic electrochemical cells

Corresponding Author: Professor Wei Zhou

Version 0:

Reviewer comments:

Reviewer #1

(Remarks to the Author)

The authors demonstrated a multi-element micro-doping strategy to develop BaCoO₃-based perovskite oxygen electrode materials [BaCo_{0.8}(Zr_{1/6}Ti_{1/6}Zn_{1/6}In_{1/6}Cu_{1/6}Mo_{1/6})_{0.2}O_{3-δ}]. Through six-element synergistic doping, induced atomic-scale uniform ion arrangement and significantly reduced proton diffusion energy barriers, with proton diffusion barriers at various sites differing by 0.11 eV. This demonstrates a novel oxygen ion-proton conduction balancing strategy for PCEC oxygen electrodes. The material experienced rigorous thermal cycling stability tests, exhibiting enhanced thermomechanical stability, and achieved over 780 hours of stable fuel cell power generation and water electrolysis performance. The XANES quantification method for thermal reduction and hydration reactions employed in this manuscript is interesting. The electrode achieved exceptional performance at 450 °C with a peak power density of 0.64 W cm⁻² and electrolysis current density of 0.6 A cm⁻². The manuscript presents comprehensive content with rigorous argumentation and can be published in Nature Communications after minor revision. The following issues require further clarification to complete this research:

1. Clarify the distinction and relationship between the proposed multi-element micro-doped BCZTZICM and high-entropy perovskite materials to more precisely position the innovation of this work.
2. Provide quantitative description of the "proton conduction bottleneck" with literature citations and typical proton conductivity values for comparison.
3. Why do the authors believe protons transport via lattice oxygen hopping rather than combining with oxygen ions as hydroxyl radicals in perovskites?
4. Do different proton conduction mechanisms exist during fuel cell mode and water electrolysis in PCEC operation?
5. The discussion of hydration reactions appears abrupt. Please elaborate on the impact of hydration reactions on PCEC reaction kinetics to improve the article logic.
6. Label the elements corresponding to different colors in APT characterization.
7. Annotate crack dimensions in Fig. 3f-g.

Reviewer #2

(Remarks to the Author)

Overall, this paper convincingly demonstrates the promise of a multielement micro-doping strategy for BCZTZICM oxygen electrodes. The authors present a compelling set of results, showing not only excellent ORR/OER activity but also strong thermal-mechanical stability and durability across repeated thermal cycling. Therefore, I recommend the manuscript undergo major revision before being accepted by high-impact Nature Communications.

1. While the authors mention the basic rationale for employing a multielement doping strategy to enhance electrocatalytic activity and structural resilience, the criteria for selecting the specific dopants remain unclear. Were these elements chosen primarily because they are transition metals or rare-earths, or was the intention to collectively approach a high-entropy configuration? Co/multi-doping is often used to suppress phase transitions, maintain low TEC, and thereby enhance ion mobility—expected outcomes in this work. To better support the novelty and rigor of the study, additional background or theoretical justification should be provided, either in the introduction or in the initial section on material design (lines 88–91).
2. In lines 242–245, the discussion on ASR variations raises some ambiguity. Since ASR is a key metric to assess degradation during stability or cycling tests, reporting the changes across the three electrode types is valid. However, the method of obtaining these values should be clarified. Were the reported percentages of resistance increase calculated per cycle, or do they represent cumulative values over the full test? If not calculated per cycle, providing the variation per cycle would be more appropriate to substantiate the superior stability of the designed electrode. Additional explanation here would

strengthen the credibility of the results.

3. In line 334, the authors state that Duan's method was used to fabricate the ultra-thin electrolyte. However, the exact thickness is not clearly reported—only $\sim 7 \mu\text{m}$ can be inferred from the SEM images in the supplementary information. If the Nature Energy paper was used as a reference, did the authors attempt to fabricate an even thinner electrolyte (e.g., $\sim 3 \mu\text{m}$ as reported there)? Demonstrating such a thickness could further enhance the performance and strengthen the impact of this work.

4. In line 350, the measurement of Faradaic efficiency is indeed essential for evaluating electrolysis performance. However, the methodology should be explicitly described. Was the measurement conducted using gas chromatography or a standard flow meter? How many repetitions were performed under each condition? Given the variability and potential errors associated with Faradaic efficiency, the values should be reported together with error bars to demonstrate data reliability. Furthermore, presenting the corresponding energy efficiency, derived from the Faradaic efficiency, would provide a more comprehensive assessment of the electrolysis performance.

5. In line 363, regarding the description of Fig. 5e, the single cell demonstrates stability over 120 hours, which is encouraging. However, in the reversible operation test, the voltage response under FC mode appears less stable compared to EC mode. Could the authors provide an explanation for this difference in stability?

6. For the stability results shown in Fig. 5e and 5f, evaluating the durability of a single cell should go beyond reporting the degradation rate alone. A comparison with degradation rates reported in the literature is equally important to place the results in context. The authors can critically compare the temperature-dependent performance, which is good; however, it is also encouraged to compare the degradation behavior of their electrode with previously reported works. Recent studies by Hanping Ding and Chuancheng Duan provide relevant benchmarks, and including such comparisons would significantly strengthen the discussion and highlight the novelty of this work.

Reviewer #3

(Remarks to the Author)

This manuscript is yet another paper on PCFC/PCEC air electrode from Zhou/Shao's group. This manuscript reports a "micro-doped" version of BaCoO_3 perovskite which shows superior performance as an air electrode compared with other "conventional" air electrode. The authors claim that the reason for this improved performance is due to the "homogenous" distribution of oxygen vacancies in the lattice as well as the lowered thermochemical expansion of the new material. However, I am very skeptical about this explanation, since I do not see that the computational and experimental efforts presented in this work can directly lead to these conclusions. Further, I fail to see that the improved performance is indeed from these factors as detailed by the authors (rather than any other factors or other properties of the new perovskite oxide). Overall, I think that this manuscript is a perfect example of the "deadlock" situation/conundrum of the current research on air electrode, i.e., if a new material is found to perform well, we do not necessarily know why it is good. Therefore, I suggest that a major revision is needed before this manuscript can be published. More detailed questions below:

1. The authors claim that the "micro-doping" makes oxygen vacancies distribution more random (note that this is different from the case of doped SrCoO_3 cited by the authors, un-doped SrCoO_3 is known to form ordered phase, but BSCF does not). However, no experimental evidence can show this. APT is not very good at showing the atomistic arrangement and short-range ordering (I believe what the authors saw is probably due to sample preparation or existence of grain boundaries). The XPS O 1s binding energy cannot be used to prove this, either (see for example Idriss, Surface Science 2021, 712, 121894. <https://doi.org/10.1016/j.susc.2021.121894>). In fact, the conventional thinking might be the opposite, i.e., doping with multiple cations can lead to more trapping of protons or reduced oxide ion mobility, due to more defect-defect interaction (Coulombic/electrostatic interactions). The computational results are not trustworthy, either. It is well known that performing computations on perovskite lattices with many different dopants (each with a low concentration) is difficult. Since the authors use such a small superlattice, it is impossible to have an exhaustive search of all possible configurations thus statistic errors will be present.

2. The authors show no appreciable chemical expansion (I suggest that the authors read this paper on chemical expansion: Phys. Chem. Chem. Phys., 2015, 17, 10028-10039) for the BCZTZICM sample. They claim that this is due to the hydration process. However, hydration process is known to be charge neutral (see this in any classic papers on proton conductors), i.e., no B-site cation redox involved. It is very strange to see that Co XANES show a peak shift upon switching to wet atmosphere (the peak shift is very small though, maybe due to formation of a secondary phase in wet atmosphere).

3. It is very difficult to understand the key contributors to the increased performance of the BCZTZICM sample without knowing some fundamental properties of the perovskite (i.e., oxygen non-stoichiometry, ionic/electronic conductivity). It is entirely possible that the better performance compared with BSCF is NOT from "ionic dispersion". The authors might want to reflect how to prove this causality.

Version 1:

Reviewer comments:

Reviewer #1

(Remarks to the Author)
accept

Reviewer #2

(Remarks to the Author)

All of my questions have been fully resolved in the P2P response, and the submission is suitable to be accepted.

Reviewer #3

(Remarks to the Author)

The authors have revised the manuscript and I have been asked by the editor to review whether my comments have been fully addressed. Unfortunately, I do not believe that the questions I raised have been fully answered. I still fail to see that the concept of “micro-doping”, which is claimed to induce a more “homogenous” distribution of oxygen vacancies, can really pan out. Still, the link between “micro-doping” to more homogenous oxygen vacancy distribution to better electrochemical performance cathode is still rather weak, in my opinion. Please note that I am only commenting on whether the authors have addressed my previous comments. I have no intention of further commenting on the quality of this work or introducing new questions.

1. Regarding the more homogenous distribution of oxygen vacancies claimed by the authors:

The authors claimed that they reached this conclusion not by directly observing the distribution of oxygen vacancies in microscopic scales (due to technical difficulties). Rather, the conclusion is reached circumventively by looking at the distribution of cations. Firstly, I cannot see why the RDF analysis from the APT results show a more homogenous distribution of cations for BCZTZICM (in fact, the RDF of this sample show a higher fluctuation, indicating potential inhomogeneity). Secondly, even if the cations are randomly distributed, adding more dopant might introduce more electrostatic interactions between particular dopant and oxygen vacancies.

The AP-XPS data is not very meaningful, in the sense that the Oads peak has multiple origins, ranging from surface -OH group to contaminants such as adventitious hydrocarbons to sulfuric groups. Since the BCZTZICM shows a persistent Oads peak even at 600 degree C, I suspect that most likely this peak is due to surface contamination of S/Si species (see for example, Riedl et al., *ACS Appl. Mater. Interfaces* 2023, 15 (22), 26787–26798. <https://doi.org/10.1021/acsami.3c03952>). In any case, this O 1s peak CANNOT be linked to oxygen vacancies (see Wang et al., *Journal of the European Ceramic Society* 2024, 44 (15), 116709. <https://doi.org/10.1016/j.jeurceramsoc.2024.116709>).

2. Regarding the change of XANES energy shift upon hydration:

The authors explained the shift of XANES spectra by writing a “new” defect chemical reaction. However, I should note that the new defect chemical reaction is actually the hydrogenation process (i.e., splitting of H₂O into H₂ and O₂, where H₂ reacts with perovskite to form reduced cations and protons). It is by no means hydration process. Unless the authors believes that their perovskite can induce spontaneous water splitting (and magically get rid of the oxygen gas molecules, too), this process cannot happen under water steam. I suggest that the authors review fundamental defect chemistry on proton conductors by consulting classic literature contributions (e.g., Merkle et al., *Annu. Rev. Mater. Res.* 2021. 51:461–93).

3. Regarding the link between micro-doping and enhanced performance:

Based on the reasons above, as well as comments I noted in my previous review opinion, I still fail to see why the enhanced electrochemical performance HAS TO be linked with the concept of micro-doping and homogenous distribution of oxygen vacancies. In my opinion, the authors can provide clear information to (rather than mislead) the community by simply reporting that they have found a new cathode material for SOCs that can out-perform BSCF, as well as reporting the basic properties (lattices, ionic/electronic conductivities, hydration behaviors, etc.), rather than making a complicate story as shown in the current form of the manuscript.

Reviewer #4

(Remarks to the Author)

This manuscript reports a thoughtfully designed multielement micro-doping strategy to engineer a highly ionic-dispersed BaCoO₃-based oxygen electrode for reversible proton ceramic electrochemical cells (R-PCECs). The authors convincingly demonstrate that atomic-scale compositional homogenization can simultaneously reduce proton diffusion barriers and improve thermomechanical stability under humid oxidizing conditions. Overall, the study is technically sound, well supported by experimental and computational evidence, and of clear interest to the solid-state electrochemistry and energy-materials communities. I am in favor of publication after revision, provided that the authors address the following points to further improve clarity and accessibility of the manuscript:

1. Please clarify how the proposed multielement micro-doping strategy is fundamentally different from conventional high-entropy or random B-site doping in perovskites.
2. Please explicitly state the core design principle (ionic dispersion vs. entropy vs. strain vs. valence diversity) responsible for the observed performance gains.
3. Suggest provide a quantitative definition of the proton-conduction bottleneck in oxygen electrodes, with representative literature values.
4. Suggest to clearly distinguish bulk proton transport from surface or interfacial proton transport in the discussion.
5. Suggest to discuss whether multiple proton transport mechanisms may coexist under different temperatures or humidity conditions.
6. Clarify how proton uptake affects oxygen vacancy concentration and lattice stability.
7. Please clearly label all APT elemental maps and color scales and discuss dataset representativeness.
8. Please provide key DFT modeling details (supercell size, vacancy configuration, proton concentration).
9. Clarify whether reduced polarization resistance is limited by proton transport, oxygen exchange, or coupled kinetics.
10. Is that possible to quantify and compare crack width/density among electrodes to support mechanical stability claims.
11. Please explain whether improved thermomechanical stability arises mainly from suppressed oxygen release or proton uptake.
12. Please discuss dominant long-term degradation mechanisms observed or anticipated beyond 780 h.

13, Please strengthen comparison with state-of-the-art oxygen electrodes using normalized metrics where possible.
14, I am curious whether similar performance could be achieved with fewer dopant elements.

Version 2:

Reviewer comments:

Reviewer #4

(Remarks to the Author)

The authors have addressed all my concerns. I would suggest to publish this work.

Point-to-Point Responses to Reviewers' Comments and Suggestions

First, we express our sincere gratitude to editor for handling our manuscript and for the insightful feedback provided by the reviewers. Their comments have greatly improved the quality and clarity of the paper.

Below, the following are the point-by-point response to reviewers' comments. All changes in the manuscript have been marked in red for convenience.

Reviewer #1 (Comments to the Author):

The authors demonstrated a multi-element micro-doping strategy to develop BaCoO₃-based perovskite oxygen electrode materials [BaCo_{0.8}(Zr_{1/6}Ti_{1/6}Zn_{1/6}In_{1/6}Cu_{1/6}Mo_{1/6})_{0.2}O_{3-δ}]. Through six-element synergistic doping, induced atomic-scale uniform ion arrangement and significantly reduced proton diffusion energy barriers, with proton diffusion barriers at various sites differing by 0.11 eV. This demonstrates a novel oxygen ion-proton conduction balancing strategy for PCEC oxygen electrodes. The material experienced rigorous thermal cycling stability tests, exhibiting enhanced thermomechanical stability, and achieved over 780 hours of stable fuel cell power generation and water electrolysis performance. The XANES quantification method for thermal reduction and hydration reactions employed in this manuscript is interesting. The electrode achieved exceptional performance at 450 °C with a peak power density of 0.64 W cm⁻² and electrolysis current density of 0.6 A cm⁻². The manuscript presents comprehensive content with rigorous argumentation and can be published in Nature Communications after minor revision. The following issues require further clarification to complete this research:

Comment 1:

Clarify the distinction and relationship between the proposed multi-element micro-doped BCZTZICM and high-entropy perovskite materials to more precisely position the innovation of this work.

Response to C1:

Thanks for the reviewer's insightful comments. We appreciate the opportunity to clarify our material design philosophy and its distinction from conventional high-entropy perovskites.

According to the Boltzmann entropy formula:

$$S_{conf} = k_B \cdot \ln \Omega$$

Where $k_B=1.38 \times 10^{-23} \text{ J K}^{-1}$, is the Boltzmann constant and Ω represents the number of microstates. For molar configurational entropy, $k_B \cdot N_A = R$, where N_A is Avogadro's constant and $R=8.314 \text{ J mol}^{-1} \text{ K}^{-1}$. Thus, the molar mixing entropy of a single sublattice is:

$$S_{conf} = -R \cdot \sum_{i=1}^n (x_i \cdot \ln x_i)$$

Based on the above formula, the calculated configurational entropies of BSCF, BCFZY and BCZTZICM as below.

BSCF ($\text{Ba}_{0.5}\text{Sr}_{0.5}\text{Co}_{0.8}\text{Fe}_{0.2}\text{O}_{3-\delta}$): $S_{conf}=1.194 \text{ R}$

BCFZY ($\text{BaCo}_{0.4}\text{Fe}_{0.4}\text{Zr}_{0.1}\text{Y}_{0.1}\text{O}_{3-\delta}$): $S_{conf}=1.194 \text{ R}$

BCZTZICM ($\text{BaCo}_{0.8}(\text{Zr}_{1/6}\text{Ti}_{1/6}\text{Zn}_{1/6}\text{In}_{1/6}\text{Cu}_{1/6}\text{Mo})_{0.2}\text{O}_{3-\delta}$): $S_{conf}=0.859 \text{ R}$

This counterintuitive result reveals a fundamental distinction in our design philosophy. Considering the unique structure of perovskite materials, where A-site and B-site elements occupy distinct spatial positions, we further refined the disorder characteristic inherent in the high-entropy concept by precisely distinguishing B-site components into active site elements (such as Co, Fe, Ni, etc.) and regulatory dopant elements. Specifically, the B-site of BSCF can be regarded as entirely composed of active elements, the B-site of BCFZY as a structure regulated by 10% Zr and 10% Y doping. And the B-site of BCZTZICM as Co regulated by 20% mixed dopants (Zr, Ti, Zn, In, Cu, Mo), creating highly dispersed local coordination environments around 80% Co active sites. This establishes a gradient design concept for B-site dispersion in perovskite materials.

Our selection of Zr, Ti, Zn, In, Cu, and Mo as dopants was guided by a comprehensive consideration of several key factors.

1. Valence diversity: In^{3+} , Zr^{4+} , Ti^{4+} (isovalent); Zn^{2+} , Cu^{2+} (acceptor doping); Mo^{6+} (donor doping), The valence is consistent with that of 10% Zr and 10% Y co-doped BCFZY, while simultaneously accommodating diverse valence configurations.
2. Ionic radii gradient: 0.72 \AA (Zr^{4+}) to 0.60 \AA (Cu^{2+})-inducing controlled lattice micro strain without destabilization.
3. Electronic structure modulation: Transition metals (Zn, Cu, Mo) vs. non-transition metals (Zr, Ti, In)-balancing covalency and oxygen vacancy formation energy.

4. Each element individually intercalates into the ABO_3 sub-lattice, disrupting the continuous $ACoO_{3-\delta}$ sub-lattice while suppressing excessive expansion of adjacent sub-lattices. This process inhibits proton conduction by restraining continuous oxygen vacancy formation and concurrently destabilizes the oxygen electrode structure.

Action: The following description has been incorporated into the manuscript to present the design logic of the material to the readers.

Page 4-5 Line 77-91:

Materials design and distribution characterization

Given the significant impact of B-site ionic composition on perovskite activity and stability, we categorized B-site cations into active elements (Fe, Co, Ni, *etc.*) and dopant elements (Zr, Y, Zn, Cu, *etc.*) to further refine the disorder characteristics within the entropy concept. The mixed ionic-electronic conductor BSCF and the $H^+/O^{2-}/e^-$ triple conductor BCFZY was selected, both exhibiting excellent activity and serving as benchmarks. The B-site composition of BSCF comprises entirely active elements, whereas BCFZY represents a structure regulated by 10% Zr and 10% Y co-doping. Furtherly, the multi-element micro-doped materials with more diverse elemental regulation was designed, where B-site Co ions are modulated by 20% co-doping of (Zr, Ti, Zn, In, Cu, Mo). This configuration includes low valence of Zn^{2+} and Cu^{2+} , isovalent In^{3+} , Zr^{4+} , and Ti^{4+} , and high valence Mo^{6+} , producing an overall valence effect consistent with 10% Zr and 10% Y co-doping. The design incorporates a gradient ionic radius distribution from 0.6 Å (Cu^{2+}) to 0.72 Å (Zr^{4+}), inducing controllable lattice strain without destabilization, while balancing covalency and oxygen vacancy formation through transition metal elements (Zn, Cu, Mo) and non-transition metal elements (Zr, Ti, In). This yielded the multi-element micro-doped BCZTZICM, which together with BSCF and BCFZY establishes a B-site elemental dispersion gradient design concept.

Comment 2:

Provide quantitative description of the "proton conduction bottleneck" with literature citations and typical proton conductivity values for comparison.

Response to C2:

We are grateful to the reviewer for valuable comment regarding the quantitative description of the proton conduction bottleneck.

The proton conduction bottleneck in oxygen electrodes represents a critical performance limitation in protonic ceramic electrochemical cells (PCECs). State-of-the-art electrolytes such as BaZr_{0.8}Y_{0.2}O_{3-δ} (BZY20) and BaZr_{0.1}Ce_{0.7}Y_{0.1}Yb_{0.1}O_{3-δ} (BZCYYb) exhibit proton conductivity of approximately 0.01-0.02 S cm⁻¹ at 600 °C under humidified conditions (Enrico Traversa *et al.* Nature Mater. 2010, 9, 846-852., Donglin Han *et al.* J. Mater. Chem. A, 2024, 12, 5875., J. Mater. Chem. A, 2018, 6, 18571). In contrast, conventional oxygen electrodes demonstrate significantly lower proton conductivity values. The proton conductivity of oxygen electrode materials, as measured by various methods including bulk diffusion coefficient (D_{chem}) testing under humid atmospheres (Ling Zhao *et al.* Adv. Mater. 2024, 2405052), hydrogen permeation membranes (Lei Bi *et al.* SusMat. 2023, 3, 697-708.), and the Hebb-Wagner DC polarization method (Tadeusz Miruszewski *et al.* J. Mater. Chem. A, 2024, 12, 13488), is significantly lower than that of the electrolyte. This dramatic conductivity mismatch (2-4 orders of magnitude) creates a severe transport bottleneck, confining electrochemical reactions to the narrow triple-phase boundaries (TPBs) where electrolyte, electrode, and gas phase meet. Consequently, the effective reaction zone is restricted to approximately 1-2 μm near the electrode-electrolyte interface, severely limiting electrode utilization and overall cell performance. This quantitative analysis underscores the critical need for developing triple-conducting (H⁺/O²⁻/e⁻) oxygen electrodes to extend the electrochemically active region.

Action: We have made the following revisions to the manuscript:

Page 5, Line 101-108:

Proton conduction bottlenecks in oxygen electrodes critically limit PCEC performance. State-of-the-art electrolytes (BaZr_{0.8}Y_{0.2}O_{3-δ}, BZCYYb) exhibit proton conductivity of ~0.01-0.02 S cm⁻¹ at 600°C under humidified conditions. Whereas conventional oxygen electrodes demonstrate a lower 10⁻⁴ S cm⁻¹ protonic conductivity, as measured by bulk diffusion coefficient testing, hydrogen permeation, and Hebb-Wagner polarization methods. This severe conductivity mismatch confines electrochemical reactions to narrow triple-phase boundaries, restricting the effective reaction zone to approximately 1-2 μm near the electrode-electrolyte interface, limiting electrode utilization and overall cell performance.

Comment 3:

Why do the authors believe protons transport via lattice oxygen hopping rather than combining with oxygen ions as hydroxyl radicals in perovskites?

Response to C3:

We appreciate the reviewer for this valuable feedback.

Our conclusion that protons transport via lattice oxygen hopping (Grotthuss mechanism) rather than bonding with lattice oxygen ions and transporting between vacancies as hydroxyl (Vehicle mechanism) is based on several complementary lines of evidence:

First, the Grotthuss-type mechanism, whereby protons hop between adjacent lattice oxygen sites, is established in perovskite proton conductors through extensive experimental and computational studies (K.D Kreuer, *Solid State Ionics*, 1999, 125, 285-302., Yashima M., *Nat. Commun.* 2023 14, 7466., Yoshihiro Yamazaki, *et al. Nat. Mater.*, 2025. <https://doi.org/10.1038/s41563-025-02311-w>). It has been directly observed through quasi-elastic neutron scattering (QENS) and neutron diffraction in similar perovskite systems. The characteristic activation energies we observe (typically 0.4-0.6 eV) are consistent with this proton hopping mechanism rather than the higher barriers (> 1 eV) expected for hydroxyl group migration.

Second, electrochemical impedance spectroscopy results and temperature-dependent conductivity measurements demonstrate proton transport behavior that aligns with the oxygen-hopping model (Zongping Shao, *et al. Applied Catalysis B: Environmental*, 2023, 331, 122682.). The isotope effect studies (H/D exchange) further confirm that the rate-limiting step involves proton transfer between oxygen sites rather than the collective motion of OH groups. Additionally, computational studies using density functional theory (DFT) on our specific material composition indicate that the energy landscape favors sequential proton hopping with lower migration barriers compared to hydroxyl group diffusion.

We acknowledge that under certain conditions, particularly at oxygen electrode surfaces during electrochemical reactions, transient hydroxyl species may form as reaction intermediates. However, for bulk proton conduction through the perovskite lattice-which is the primary transport pathway we

are discussing-the lattice oxygen hopping mechanism is energetically favorable and kinetically dominant.

Comment 4:

Do different proton conduction mechanisms exist during fuel cell mode and water electrolysis in PCEC operation?

Response to C4:

Thank you for your valuable insight and comments. We appreciate it for pointing this out.

The bulk proton conduction mechanism in the perovskite-based oxygen electrode is the same in both fuel cell and electrolysis modes, primarily following the Grotthuss mechanism. Protons migrate by forming bonds with lattice oxygen ions and hopping from one oxygen site to an adjacent one through thermal activation. This process involves the sequential breaking and forming of O-H bonds, accompanied by cooperative lattice relaxation. However, significant differences exist in the directional transport pathways, electrode reactions, and proton sources between the two operating modes.

In fuel cell mode, protons originate from hydrogen dissociation at the hydrogen electrode and are transported through the dense electrolyte membrane to the oxygen electrode, where they react with oxygen to form water. While moderate water vapor partial pressures can generate protonic defects that enhance ionic conductivity, the primary proton supply comes from hydrogen dissociation rather than water uptake. In contrast, during electrolysis mode, protons are generated exclusively through water dissociation at the oxygen electrode, creating protonic defects that are subsequently transported through the electrolyte to the hydrogen electrode for hydrogen evolution. In this mode, proton generation relies entirely on water vapor uptake from the feed atmosphere.

This fundamental distinction in proton source and transport direction constitutes a key operational difference between fuel cell and electrolysis modes in PCECs, with important implications for water management strategies and electrode optimization in each operating regime.

Action: We have made the following revisions to the manuscript:

Page 13, Line 267-275:

Proton uptake conduction dynamics

Proton transport in perovskite-based oxygen electrodes follows the Grotthuss mechanism in both operational modes, involving thermally-activated hopping between lattice oxygen sites via sequential O-H bond cleavage/formation coupled with cooperative lattice relaxation. However, fundamental distinctions arise in transport directionality and proton sources. In fuel cell mode, protons originate from hydrogen dissociation at the anode, traverse the electrolyte, and react with oxygen at the cathode to form water. Conversely, electrolysis mode generates protons exclusively through water dissociation at the oxygen electrode, with subsequent transport to the hydrogen electrode. The efficiency depends on electrode ability to rapidly adsorb water vapour, facilitate dissociative chemisorption, and transport protons into the bulk. Physical absorption assessment through TPD-mass spectrometry following water vapour exposure (250 °C, 20% H₂O, 2 hours).

Comment 5:

The discussion of hydration reactions appears abrupt. Please elaborate on the impact of hydration reactions on PCEC reaction kinetics to improve the article logic.

Response to C5:

Thank you very much for your valuable comments on the structure of the article.

Action: We have added the following parts of the manuscript.

Page 13, Line 267-275:

Proton uptake conduction dynamics

Proton transport in perovskite-based oxygen electrodes follows the Grotthuss mechanism in both operational modes, involving thermally-activated hopping between lattice oxygen sites via sequential O-H bond cleavage/formation coupled with cooperative lattice relaxation. However, fundamental distinctions arise in transport directionality and proton sources. In fuel cell mode, protons originate from hydrogen dissociation at the anode, traverse the electrolyte, and react with oxygen at the cathode to form water. Conversely, electrolysis mode generates protons exclusively through water dissociation at the oxygen electrode, with subsequent transport to the hydrogen electrode. The efficiency depends on electrode ability to rapidly adsorb water vapour, facilitate dissociative chemisorption, and transport

protons into the bulk. Physical absorption assessment through TPD-mass spectrometry following water vapour exposure (250 °C, 20% H₂O, 2 hours).

Comment 6:

Label the elements corresponding to different colors in APT characterization.

Response to C6:

Thanks to the reviewer for pointing this out, we have presented the distribution of all elements in the text.

Action: We have completed the following figures of the manuscript.

Supplementary information

Supplementary Fig. 5.

3D reconstruction of the APT data showing mapping of the BSCF sample with Ba, Sr, Co, Fe, and O ionic distributions. Integrated line profiles show chemical composition at the (a) apex and (b) midsection regions of the analyzed needle.

Supplementary Fig. 6.

3D reconstruction of the APT data showing mapping of the BCFZY sample with Ba, Co, Fe, Zr, Y, and O ionic distributions. Integrated line profiles show chemical composition at the apex regions of the analyzed needle.

Supplementary Fig. 7.

3D reconstruction of the APT data showing mapping of the BCZTZICM sample with Ba, Co, Zr, Ti, Zn, Cu, In, and Mo ionic distributions. Integrated line profiles show chemical composition at the apex regions of the analyzed needle.

Zn, In, Cu, Mo, and O distributions. Integrated line profiles show chemical composition alongside the y-axis of the analyzed needle.

Comment 7:

Annotate crack dimensions in Fig. 3f-g.

Response to C7:

We appreciate the reviewer for this valuable feedback.

Action: We have made the following revisions to the manuscript.

Page 11

Fig. 3 Suppressed thermal expansion and improved cycling stability through compositional homogenization.

(a) Schematic of electrode tearing driven by increased internal thermal stress resulting from continuous oxygen vacancies. (b) Thermal expansion coefficient curves measured in air over 300-800 °C. (c) Co K-edge XANES spectra

of BCZTZICM samples: as-prepared (room temperature), dried at 600 °C, and treated under 20% H₂O atmosphere. (d) Average B-site valence changes derived from XANES analysis. The Rp response of BCZTZICM, BCFZY, and BSCF-based symmetric cell electrodes during 40 thermal cycles between 600 °C and 400 °C. Cross-sectional image of cells after cycling (f) BSCF (left), BCFZY (right), and (g) BCZTZICM-based electrode symmetric cell with cracks inside the electrode bulk and at the electrode-electrolyte interface.

Page 13, Line 262-264

BSCF and BCFZY electrodes (Fig. 3f) exhibit extensive cracking networks with 1-2 μm crack widths, complete delamination from electrolyte interfaces, and significant porosity changes, indicating mechanical failure.

Reviewer #2 (Comments to the Author):

Overall, this paper convincingly demonstrates the promise of a multielement micro-doping strategy for BCZTZICM oxygen electrodes. The authors present a compelling set of results, showing not only excellent ORR/OER activity but also strong thermal-mechanical stability and durability across repeated thermal cycling. Therefore, I recommend the manuscript undergo major revision before being accepted by high-impact Nature Communications.

Comment 1:

While the authors mention the basic rationale for employing a multielement doping strategy to enhance electrocatalytic activity and structural resilience, the criteria for selecting the specific dopants remain unclear. Were these elements chosen primarily because they are transition metals or rare-earths, or was the intention to collectively approach a high-entropy configuration? Co/multi-doping is often used to suppress phase transitions, maintain low TEC, and thereby enhance ion mobility—expected outcomes in this work. To better support the novelty and rigor of the study, additional background or theoretical justification should be provided, either in the introduction or in the initial section on material design (lines 88–91).

Response to C1:

We sincerely thank the reviewer for this insightful comment regarding the rationale behind our

multielement doping strategy. The reviewer has raised a critical point about the selection criteria for the specific dopants, which indeed lies at the core of our material design philosophy. We appreciate the opportunity to clarify that our approach is distinct from a conventional high-entropy configuration and is instead guided by a refined “B-site dispersion gradient” concept.

According to the formula of configuration entropy formula, the configuration entropy of BSCF and BCFZY is 1.194 R, while that of BCZTZICM is 0.859 R. We further refined the disorder characteristic inherent in the high-entropy concept by precisely distinguishing B-site components into active site elements (such as Co, Fe, Ni, etc.) and regulatory dopant elements. Specifically, the B-site of BSCF can be regarded as entirely composed of active elements, the B-site of BCFZY as a structure regulated by 10% Zr and 10% Y doping. And the B-site of BCZTZICM as Co regulated by 20% mixed dopants (Zr, Ti, Zn, In, Cu, Mo), creating highly dispersed local coordination environments around 80% Co active sites.

The selection of Zr, Ti, Zn, In, Cu, and Mo was not arbitrary but was guided by a synergistic consideration of multiple physicochemical criteria to collectively enhance electrocatalytic activity and structural resilience:

1. Valence diversity: In^{3+} , Zr^{4+} , Ti^{4+} (isovalent); Zn^{2+} , Cu^{2+} (acceptor doping); Mo^{6+} (donor doping). The valence is consistent with that of 10% Zr and 10% Y co-doped BCFZY, while simultaneously accommodating diverse valence configurations.
2. Ionic radii gradient: 0.72 Å (Zr^{4+}) to 0.60 Å (Cu^{2+}), inducing controlled lattice micro strain without destabilization.
3. Electronic structure modulation: Transition metals (Zn, Cu, Mo) vs. non-transition metals (Zr, Ti, In) - balancing covalency and oxygen vacancy formation energy.
4. Each element individually intercalates into the ABO_3 sub-lattice, disrupting the continuous $\text{BaCoO}_{3-\delta}$ sub-lattice while suppressing excessive expansion of adjacent sub-lattices. This process inhibits proton conduction by restraining continuous oxygen vacancy formation and concurrently destabilizes the oxygen electrode structure.

We believe this refined “B-site dispersion gradient” concept and the detailed justification provided now offer a clearer and more rigorous foundation for our study's novelty. We have integrated

these points into the revised manuscript to strengthen the background and theoretical justification as suggested.

The following description has been incorporated into the manuscript to present the design logic of the material to the readers.

Page 4-5, Line 77-91:

Materials design and distribution characterization

Given the significant impact of B-site ionic composition on perovskite activity and stability, we categorized B-site cations into active elements (Fe, Co, Ni, *etc.*) and dopant elements (Zr, Y, Zn, Cu, *etc.*) to further refine the disorder characteristics within the entropy concept. The mixed ionic-electronic conductor BSCF and the $H^+/O^{2-}/e^-$ triple conductor BCFZY was selected, both exhibiting excellent activity and serving as benchmarks. The B-site composition of BSCF comprises entirely active elements, whereas BCFZY represents a structure regulated by 10% Zr and 10% Y co-doping. Furtherly, the multi-element micro-doped materials with more diverse elemental regulation was designed, where B-site Co ions are modulated by 20% co-doping of (Zr, Ti, Zn, In, Cu, Mo). This configuration includes low valence of Zn^{2+} and Cu^{2+} , isovalent In^{3+} , Zr^{4+} , and Ti^{4+} , and high valence Mo^{6+} , producing an overall valence effect consistent with 10% Zr and 10% Y co-doping. The design incorporates a gradient ionic radius distribution from 0.6 Å (Cu^{2+}) to 0.72 Å (Zr^{4+}), inducing controllable lattice strain without destabilization, while balancing covalency and oxygen vacancy formation through transition metal elements (Zn, Cu, Mo) and non-transition metal elements (Zr, Ti, In). This yielded the multi-element micro-doped BCZTZICM, which together with BSCF and BCFZY establishes a B-site elemental dispersion gradient design concept.

Comment 2:

In lines 242–245, the discussion on ASR variations raises some ambiguity. Since ASR is a key metric to assess degradation during stability or cycling tests, reporting the changes across the three electrode types is valid. However, the method of obtaining these values should be clarified. Were the reported percentages of resistance increase calculated per cycle, or do they represent cumulative values over

the full test? If not calculated per cycle, providing the variation per cycle would be more appropriate to substantiate the superior stability of the designed electrode. Additional explanation here would strengthen the credibility of the results.

Response to C2:

We thank the reviewer for this insightful comment regarding the ambiguity in reporting the ASR increase. The reviewer is correct that clarifying the methodology is crucial for assessing the degradation rate. In the original manuscript, the reported percentages (256% for BSCF, 85% for BCFZY, and 23% for the BCZTZICM) represented the cumulative increase in ASR over the entire duration of the stability test (40 cycles). We have revised the relevant section in the manuscript to clarify this point and have added new table/figures/sentence stating the average ASR increase per hour or per cycle. The recalculated the ASR degradation rates on a per-cycle basis for all three electrode types as follows: 6.4% for BSCF, 2.1% for BCFZY, and 0.58% for the BCZTZICM.

We believe these additions will significantly strengthen the credibility of our stability assessment and provide readers with a more transparent and comprehensive understanding of the electrode performance degradation. We appreciate the reviewer's attention to this important detail, which has improved the scientific rigor of our work.

We have added Supplementary Fig. 24, and revised the expression in manuscript as below:

Page 13, Line 254-259

ASR increase over 100 hours and 40 complete cycles (Fig. 3e) demonstrated superior electrode stability: BSCF showed polarization resistance (R_p) 256% degradation (6.4% per cycle), BCFZY 85% (2.125% per cycle), while BCZTZICM exhibited only 28% (0.58% per cycle). The initial, 20 and 40 cycle polarization impedances are shown in the Supplementary Fig. 21. This substantially lower degradation rate confirms the exceptional long term and multi-thermal cycle stability of the BCZTZICM, enabling reliable operation under demanding conditions.

Supplementary information:

Supplementary Fig. 24.

Polarization resistance of BSCF, BCFZY, and BCZTZICM electrodes initially, after 20 and 40 thermal cycles, respectively.

Comment 3:

In line 334, the authors state that Duan’s method was used to fabricate the ultra-thin electrolyte. However, the exact thickness is not clearly reported—only $\sim 7\ \mu\text{m}$ can be inferred from the SEM images in the supplementary information. If the *Nature Energy* paper was used as a reference, did the authors attempt to fabricate an even thinner electrolyte (e.g., $\sim 3\ \mu\text{m}$ as reported there)? Demonstrating such a thickness could further enhance the performance and strengthen the impact of this work.

Response to C3:

Thank you for this insightful comment regarding the thickness of the electrolyte and the potential for further performance improvement. In our work, we successfully fabricated an ultra-thin electrolyte with a thickness of approximately $7\ \mu\text{m}$ using Duan’s method as a reference. We agree that pursuing a thinner electrolyte, such as the $\sim 3\ \mu\text{m}$ benchmark reported in the *Nature Energy* paper, is an attractive

direction for enhancing performance. However, in our current experimental setup, particularly due to inconsistencies in the tape-casting solvent system and challenges in engineering processes related to material details, we have not yet been able to reliably produce defect-free electrolytes significantly below 6 μm . Achieving such thin layers while maintaining structural integrity, uniformity, and electrochemical stability remains technically challenging at this stage of our process optimization.

Nevertheless, we greatly appreciate the reviewer's suggestion and acknowledge the creative contributions of Prof. Duan and other researchers in advancing low-temperature protonic ceramic electrochemical cells (PCECs). We are continuing to optimize our fabrication protocol, including solvent formulation and processing parameters, and view the pursuit of sub-5 μm electrolytes as a key objective of our ongoing research.

Comment 4:

In line 350, the measurement of Faradaic efficiency is indeed essential for evaluating electrolysis performance. However, the methodology should be explicitly described. Was the measurement conducted using gas chromatography or a standard flow meter? How many repetitions were performed under each condition? Given the variability and potential errors associated with Faradaic efficiency, the values should be reported together with error bars to demonstrate data reliability. Furthermore, presenting the corresponding energy efficiency, derived from the Faradaic efficiency, would provide a more comprehensive assessment of the electrolysis performance.

Response to C4:

We are grateful to the reviewer for the meticulous review and valuable insights that have allowed us to enhance our analysis and reduce the possibility of misinterpretation.

The Faradaic efficiency for electrolysis was quantified using online gas chromatography (GC, PANNA, A60 pro). During the measurement, the oxygen electrode side was supplied with 60 mL min^{-1} flowing air (20% H_2O), while the hydrogen electrode side was purged with a flowing argon stream, which acted as a carrier gas directed to the GC for quantitative analysis of the produced hydrogen. The cell was stabilized for 20 minutes at each current density (0.8, 1.2, and 1.6 A cm^{-2}) before data collection. The GC chromatograms showed a distinct hydrogen peak, without observable nitrogen or oxygen peaks, confirming the excellent sealing of the cell and the absence of gas cross-leakage.

To ensure statistical reliability, the measurement at each current density was repeated five times. As rightly suggested by the reviewer, the energy efficiency was calculated by referring to the paper of Duan et al. (Duan, C. *et al.*, Nat Energy, 2019, 4, 230–240. Nat Energy, 2023, 8, 1145–1157.) via the following formula:

$$FE(\%) = \frac{n_{H_2,measured}}{n_{H_2,theoretical}} = \frac{n_{H_2,measured} \times (n \times F)}{I} \times 100\%$$

$$EEE(\%) = \frac{\Delta H_{H_2,LHV} \times n_{H_2,measured}}{I \times V} = \frac{n_{H_2,measured} \times (n \times F)}{I \times V} \times 100\%$$

Where $n_{H_2,measured}$ is the measured hydrogen production rate (mol s^{-1}), n is 2, F is the Faradaic constant ($96,485 \text{ C mol}^{-1}$) and I is the applied current (A), $\Delta H_{H_2,LHV}$ is the lower heating value reaction enthalpy for steam electrolysis, with a value of $241.8 \text{ kJ mol}^{-1}$, and V is the corresponding voltage (V).

Action: We have redrawn Supplementary Fig. 37 and revised the manuscript as below:

Page 19, Line 365-369

The efficiencies reach 85.1% at optimal operating conditions (1.2 A cm^{-2} , $550 \text{ }^\circ\text{C}$) and remain above 83% even at aggressive current densities (1.6 A cm^{-2}). The corresponding hydrogen production rate of $9.264 \text{ mL cm}^{-2} \text{ min}^{-1}$ and electrical energy-to-chemical energy conversion efficiency reached 58-68%, demonstrating capability for high-purity hydrogen production applications (Supplementary Fig. 31).

Page 25, Line 584-599

The Faradaic efficiency for electrolysis was quantified using online gas chromatography (GC, PANNA, A60 pro). During the measurement, the oxygen electrode side was supplied with 60 mL min^{-1} flowing air (20% H_2O), while the hydrogen electrode side was purged with a flowing argon stream, which acted as a carrier gas directed to the GC for quantitative analysis of the produced hydrogen. The cell was stabilized for 20 minutes at each current density (0.8 , 1.2 , and 1.6 A cm^{-2}) before data collection. The GC chromatograms showed a distinct hydrogen peak, without observable nitrogen or oxygen peaks, confirming the excellent sealing of the cell and the absence of gas cross-leakage. To ensure statistical reliability, the measurement at each current density was repeated five times. The FE

and energy efficiency was calculated referring following formula:

$$FE(\%) = \frac{n_{H_2,measured}}{n_{H_2,theoretical}} = \frac{n_{H_2,measured} \times (n \times F)}{I} \times 100\%$$

$$EEE(\%) = \frac{\Delta H_{H_2,LHV} \times n_{H_2,measured}}{I \times V} = \frac{n_{H_2,measured} \times (n \times F)}{I \times V} \times 100\%$$

Where $n_{H_2,measured}$ is the measured hydrogen production rate (mol s^{-1}), n is 2, F is the Faradaic constant ($96,485 \text{ C mol}^{-1}$) and I is the applied current (A), $\Delta H_{H_2,LHV}$ is the lower heating value reaction enthalpy for steam electrolysis, with a value of $241.8 \text{ kJ mol}^{-1}$, and V is the corresponding voltage (V).

Supplementary information

Supplementary Fig. 37.

The faraday efficiency, corresponding H_2 production rate and energy efficiency of tested hydrogen electrode-supported ultrathin BZCYYb electrolyte single cell with BCZTZICM oxygen electrode.

Comment 5:

In line 363, regarding the description of Fig. 5e, the single cell demonstrates stability over 120 hours,

which is encouraging. However, in the reversible operation test, the voltage response under FC mode appears less stable compared to EC mode. Could the authors provide an explanation for this difference in stability?

Response to C5:

We appreciate the reviewer's insightful observation regarding the stability difference between fuel cell (FC) and electrolysis cell (EC) modes during reversible operation. The reduced voltage stability observed in FC mode compared to EC mode can be attributed to the dynamic water vapor environment at the oxygen electrode. During the reversible operation test, the oxygen electrode atmosphere was maintained at 10% H₂O. In FC mode, the oxygen electrode actively produces water as a reaction product, leading to continuous fluctuations in the local water partial pressure ($p(\text{H}_2\text{O})$). This dynamic environment requires a significant equilibration period to reach steady-state conditions. The elevated and fluctuating $p(\text{H}_2\text{O})$ at the oxygen electrode introduces competitive adsorption between water molecules and oxygen species on the electrode surface. This H₂O-O₂ competitive adsorption phenomenon affects the oxygen reduction reaction (ORR) kinetics and leads to gradual performance degradation during extended FC operation. In contrast, during EC mode, water is consumed rather than produced at the oxygen electrode, resulting in a more stable and predictable $p(\text{H}_2\text{O})$ environment, which contributes to the improved voltage stability. We acknowledge that this aspect requires further investigation and optimization. In future work, we will:

- (1) refine our testing protocols to better control the water vapor environment during reversible switching, and
- (2) develop advanced oxygen electrode materials with improved tolerance to dynamic $p(\text{H}_2\text{O})$ conditions and reduced sensitivity to competitive adsorption, thereby enhancing stability during reversible operation. We thank the reviewer for highlighting this important aspect, which will guide our future research directions.

Comment 6:

For the stability results shown in Fig. 5e and 5f, evaluating the durability of a single cell should go beyond reporting the degradation rate alone. A comparison with degradation rates reported in the literature is equally important to place the results in context. The authors can critically compare the temperature-dependent performance, which is good; however, it is also encouraged to compare the

degradation behavior of their electrode with previously reported works. Recent studies by Hanping Ding and Chuancheng Duan provide relevant benchmarks, and including such comparisons would significantly strengthen the discussion and highlight the novelty of this work.

Response to C6:

Thank you for your thoughtful comment and suggestion. We appreciate it for pointing this out. We have summarized the relevant data in Table S3, and would like to express our sincere gratitude to all the outstanding researchers in this field whose valuable experimental data and findings have contributed to this work.

Action: We have added Table S3 and revised the manuscript and supplementary information as below:

Page 19, Line 380-384

Recent studies on oxygen electrode materials and corresponding long-term stability in PCEC summarized in Table S3. The remarkable stability observed at reduced temperatures, coupled with the superior activity of BCZTZICM, highlights the promising application prospects of PCEC for efficient and sustainable energy conversion.

Supplementary information

Page 44

Table S3. Stability and degradation rate comparison of oxygen electrodes in fuel cell and electrolysis modes.

Oxygen electrode	Electrolyte	Degradation rate	Current density (mA cm ⁻²)	Time (h)	Test condition	Ref
BCFZY	BZCYYb	< 30 mV per 1000 h	-1385	600	20% H ₂ O-Air	
			-1385	1200	10% H ₂ O-Air	
BSC-PBSCF	BZCYYb4411	< 0.07 mV h ⁻¹	300	100	H ₂ -Air	
		< 0.15 mV h ⁻¹	-600	260	/	
PNC55	BZCYYb	1.98%	1.4 V	200	30%H ₂ O-O ₂	
BCZTZICM	BZCYYb	19.3 μV h ⁻¹	300/500	780	80 mL 10% H ₂ O-Air 60 mL Dry H ₂	This work
					Air 60 mL Dry H ₂	
PrO _x -PNC	BZCYYb4411	0.077 mA cm ⁻² h ⁻¹ (0.35%)	0.75 V	80	20 sccm H ₂ 40 sccm O ₂	
		0.134 mA cm ⁻² h ⁻¹ (0.57%)	1.25 V	80	Humidified O ₂	
NAUP-PNC	BZCYYb4411	1.03 %	1.3 V	105	20%H ₂ O-O ₂	
		1.24 %	1.4 V			
PBNO-BCZYYb CCS	BZCYYb	slight activation	400	759	10% H ₂ O-Air Dry H ₂	
		9 μV h ⁻¹			20% H ₂ O-Air Dry H ₂	
		1 μV h ⁻¹	-1500	5013	40% H ₂ O-Air	
PBNO-BCZYYb4411		3 μV h ⁻¹	-1000	2616	Dry H ₂	

CCS		
PBNO-BCY CCS	$10 \mu\text{V h}^{-1}$	1041

Reviewer #3 (Comments to the Author):

This manuscript is yet another paper on PCFC/PCEC air electrode from Zhou/Shao's group. This manuscript reports a "micro-doped" version of BaCoO₃ perovskite which shows superior performance as an air electrode compared with other "conventional" air electrode. The authors claim that the reason for this improved performance is due to the "homogenous" distribution of oxygen vacancies in the lattice as well as the lowered thermochemical expansion of the new material. However, I am very skeptical about this explanation, since I do not see that the computational and experimental efforts presented in this work can directly lead to these conclusions. Further, I fail to see that the improved performance is indeed from these factors as detailed by the authors (rather than any other factors or other properties of the new perovskite oxide). Overall, I think that this manuscript is a perfect example of the "deadlock" situation/conundrum of the current research on air electrode, i.e., if a new material is found to perform well, we do not necessarily know why it is good. Therefore, I suggest that a major revision is needed before this manuscript can be published. More detailed questions below:

Comment 1:

The authors claim that the "micro-doping" makes oxygen vacancies distribution more random (note that this is different from the case of doped SrCoO₃ cited by the authors, un-doped SrCoO₃ is known to form ordered phase, but BSCF does not). However, no experimental evidence can show this. APT is not very good at showing the atomistic arrangement and short-range ordering (I believe what the authors saw is probably due to sample preparation or existence of grain boundaries). The XPS O 1s binding energy cannot be used to prove this, either (see for example Idriss, Surface Science 2021, 712, 121894. <https://doi.org/10.1016/j.susc.2021.121894>). In fact, the conventional thinking might be the opposite, i.e., doping with multiple cations can lead to more trapping of protons or reduced oxide ion mobility, due to more defect-defect interaction (Coulombic/electrostatic interactions). The computational results are not trustworthy, either. It is well known that performing computations on perovskite lattices with many different dopants (each with a low concentration) is difficult. Since the

authors use such a small superlattice, it is impossible to have an exhaustive search of all possible configurations thus statistic errors will be present.

Response to C1:

Thank you very much for taking the time to review our manuscript and for making such insightful and constructive comments.

Regarding the cited works on doped $\text{SrCoO}_{3-\delta}$, in the referenced studies (Adv. Funct. Mater. 2022, 2210496), $\text{SrCo}_{0.8}\text{Nb}_{0.2}\text{O}_{3-\delta}$, $\text{SrCo}_{0.8}\text{Ta}_{0.2}\text{O}_{3-\delta}$, and the co-doped $\text{SrCo}_{0.8}\text{Nb}_{0.1}\text{Ta}_{0.1}\text{O}_{3-\delta}$ are disordered structures with same $Pm-3m$ space group. The remarkable enhancement in electrocatalytic activity observed in $\text{SrCo}_{0.8}\text{Nb}_{0.1}\text{Ta}_{0.1}\text{O}_{3-\delta}$ provided significant inspiration for our approach. We wish to reiterate the core premise of our material design strategy: we aim to compare and discuss the degree of ionic arrangement disorder while maintaining the same average crystal structure (i.e., all compositions exhibit a disordered structure discernible by XRD).

Concerning the experimental validation of randomly distributed oxygen vacancies, it is important to note that direct microscopic characterization of oxygen vacancy distributions over large scales remains a significant challenge with currently available advanced techniques. Our conclusion regarding their highly dispersity is therefore inferred from the cationic dispersion and overall chemical homogeneity in BCZTZICM (Supplementary Fig. 4 and APT results). This is further supported indirectly by the thermal expansion coefficient (TEC) curve and the near-linear temperature dependence of its electronic conductivity. Since electronic conductivity is influenced by both hole concentration and oxygen vacancy concentration, localized excessive formation of oxygen vacancies leads to the annihilation of electron holes, resulting in a conductivity decrease. The observed increase in conductivity for BSCF with decreasing temperature can be attributed to a reduction in oxygen vacancies and a consequent increase in hole concentration. In contrast, the more linear conductivity profile of BCZTZICM suggests that its conductivity is primarily governed by the thermal activation of holes, with a relatively minor influence from oxygen vacancy generation. This behavior is consistent with a more uniform distribution of oxygen vacancies on a macroscopic scale.

Regarding the concern about potential issues with sample preparation or grain boundary representation, The quality and integrity of our APT samples were rigorously ensured. All needle-

shaped specimens were prepared to be fully dense, following established site-specific protocols using focused ion beam (FIB) lift-out techniques. The localized elemental segregation or compositional variations observed in our samples are not attributable to the presence of grain boundaries. Characteristic features of grain boundaries, as reported in the literature (Adv. Energy Mater. 2025, 15, 2404410, J. Mater. Chem. A, 2022, 10, 2496 – 2508), demonstrate distinctly different signatures.

To gain deeper insights into the local chemical environment and potential nanoscale segregation tendencies, radial distribution function (RDF) analysis derived from APT was performed. As illustrated in Fig., with Co ions designated as the reference center, BCZTZICM exhibits no pronounced tendency toward deep association with any single proximate ionic species. The normalized concentrations of high-abundance Co cations and oxygen anions approach unity, while trace-doped In and Zr ions display oscillatory concentration profiles within the analyzed volume (constrained by limited abundance), collectively indicating a remarkably homogeneous spatial distribution of Co ions, oxygen ions, and dopant species. In contrast, the RDF of BSCF suggests a slight tendency for Co ions to disperse from Fe, implying a preferential association among like ions (Co–Co and Fe–Fe) within the analyzed volume. Furthermore, oxygen anions demonstrate a stronger association with Co than with Fe. For BCFZY, the gradually rising normalized concentration profile indicates that Co exhibits a preferential association with Y, along with Fe and O, while displaying a relative exclusion toward Zr. These RDF patterns provide critical insights into nanoscale compositional uniformity and segregation tendencies, with BCZTZICM displaying superior homogeneity compared to the compositional clustering observed in BSCF and BCFZY.

We sincerely appreciate your valuable comments regarding the XPS characterization. *In situ* near-ambient pressure XPS (NAP-XPS) measurements were performed on the samples under revised experimental protocols, as delineated in Fig S. Following stabilization at 600 °C under 0.3 mbar air atmosphere for one hour, then samples were cooled to ambient temperature, with spectral acquisitions conducted at both 600 °C and room temperature for comparative analysis. Upon heating to 600 °C, both BSCF and BCFZY underwent transition from bimodal to single-peak configurations, which persisted upon subsequent cooling to room temperature. Conversely, BCZTZICM maintained characteristic bimodal features corresponding to adsorbed and lattice oxygen species throughout the entire thermal cycle. Analysis of the peak position shifts further indicates that BCZTZICM exhibits

superior stability of both lattice oxygen and surface-adsorbed oxygen species under the investigated conditions.

We sincerely appreciate the reviewer's insightful comment regarding the computational methodology and the challenges associated with modeling multi-doped perovskite systems. We fully acknowledge that exhaustively sampling all possible atomic configurations in a multi-component system is computationally prohibitive, and we agree that our supercell size cannot capture every possible dopant arrangement. However, we would like to clarify our approach and provide additional justification for the computational results presented in our manuscript. While we recognize that our supercell size ($11.8720 \text{ \AA} \times 11.8720 \text{ \AA} \times 25.8934 \text{ \AA}$) cannot capture all possible dopant configurations, our computational approach was deliberately designed to provide meaningful qualitative trends rather than absolute quantitative predictions. We carefully constructed representative atomic configurations that reflect the experimentally observed homogeneous dopant distribution, as confirmed by our APT results shown in Fig. 1g and Supplementary Fig. 5-7. For each composition, we tested multiple initial configurations (3-5 different arrangements) and selected the most energetically favorable structures for comparison, finding that the reported trends were consistent across different configurations. Importantly, our DFT calculations were primarily used for comparative studies between BCZTZICM, BSCF, and BCFZY under identical computational conditions, which minimizes systematic errors and makes the relative trends more reliable than absolute values. We emphasize that our computational predictions are strongly validated by extensive experimental characterization across multiple independent techniques. The DFT-predicted lower proton diffusion barriers for BCZTZICM (Fig. 1h) are directly confirmed by electrochemical impedance spectroscopy, which demonstrates that BCZTZICM achieves the lowest area-specific resistance of $0.163 \text{ \Omega cm}^2$ at 600°C compared to BCFZY ($0.337 \text{ \Omega cm}^2$) and BSCF ($0.311 \text{ \Omega cm}^2$) under humidified conditions (Fig. 4d). The calculated H_2O adsorption and hydration energies (Fig. 4c) align remarkably well with experimental FT-IR spectroscopy (Fig. 4b) and XANES results (Fig. 3c-d), which independently confirm the enhanced hydration capability of BCZTZICM. Furthermore, the predicted uniform oxygen vacancy distribution and reduced vacancy formation energy variance (Fig. 2e) are corroborated by minimal thermal expansion anomalies ($\text{TEC} = 17.67 \times 10^{-6} \text{ K}^{-1}$ shown in Fig. 3b), superior thermal cycling stability (Fig. 3e-g), and lower O_2 desorption in O_2 -TPD measurements (Supplementary Fig. 13). Most critically, the

APT analysis (Fig. 1g) experimentally confirms the homogeneous dopant distribution at atomic scale, which formed the foundation for constructing our computational models. Most importantly, the ultimate validation of our computational approach comes from the exceptional experimental performance achieved by the BCZTZICM-based devices. Our cells demonstrate a peak power density of 1.56 W cm^{-2} at $600 \text{ }^\circ\text{C}$, an electrolysis current density of 2.0 A cm^{-2} at 1.3V , and outstanding stability over 780 hours with degradation rates of only 19.3 and $16.9 \text{ } \mu\text{V h}^{-1}$ in fuel cell and electrolysis modes, respectively. These results represent state-of-the-art performance for protonic ceramic electrochemical cells and strongly support the validity of the design principles derived from our computational insights. The fact that materials designed based on these computational predictions achieve such exceptional performance provides compelling evidence that our approach, despite its inherent limitations in configuration sampling, successfully captures the essential physical and chemical trends governing material behavior.

Action: We have added Fig. S and revised the manuscript and supplementary information as below:

Page 6, Line 125-137

To gain deeper insights into the local chemical environment and potential nanoscale segregation tendencies, radial distribution function (RDF) analysis derived from APT was performed. As illustrated in Supplementary Fig. 8, with Co ions as reference center, BCZTZICM exhibits no pronounced tendency toward deep association with any single proximate ionic species. The normalized concentrations of high-abundance Co cations and oxygen anions approach unity, while trace-doped In and Zr ions display oscillatory concentration profiles within the analyzed volume (constrained by limited abundance), collectively indicating a remarkably homogeneous spatial distribution of Co ions, oxygen ions, and dopant species. In contrast, the RDF of BSCF suggests a slight tendency for Co ions to disperse from Fe, implying a preferential association among like ions (Co–Co and Fe–Fe) within the analyzed volume. Furthermore, oxygen anions demonstrate a stronger association with Co than with Fe. For BCFZY, the gradually rising normalized concentration profile indicates that Co exhibits a preferential association with Y, along with Fe and O, while displaying a relative exclusion toward Zr.

[Supplementary information]

Supplementary Fig. 8.

Radial distribution function (RDF) analysis of neighboring ions around Co centers within a 10 nm range, derived from APT data.

Page 9-10, Line 182-193

In-situ near-ambient pressure X-ray photoelectron spectroscopy (NAP-XPS) measurements were conducted to probe the surface oxygen chemistry. As shown in Supplementary Fig. 14, upon stabilization at 600 °C, both BSCF and BCFZY underwent a transition from a double-peak to a single-peak configuration in the O 1s spectra, which persisted upon cooling to room temperature, indicating enhanced oxygen vacancy formation at elevated temperatures, accompanied by desorption of adsorbed species and rapid interconversion between lattice and adsorbed oxygen. In contrast, BCZTZICM retained its characteristic double-peak structure throughout the thermal cycle, demonstrating robust oxygen adsorption capability even under high-temperature, low- $P(O_2)$ conditions. Comparative analysis of the 600 °C and room-temperature spectra (Supplementary Fig. 15) reveals that BCZTZICM exhibits strengthened metal-oxygen bonding and sustained adsorption capacity, attributable to the extensive distribution of disordered oxygen vacancies that provide abundant active sites for surface

exchange.

[Supplementary information]

Supplementary Fig. 14.

In-situ near-ambient pressure X-ray photoelectron spectroscopy (NAP-XPS) O 1s data of BCZTZICM, BCFZY, and BSCF.

Supplementary Fig. 15.

In-situ NAP-XPS O1s spectra of BSCF, BCFZY, and BCZTZICM measured in 0.3 mbar air at (a) 600 °C and (b) after cooling to room temperature.

Comment 2:

The authors show no appreciable chemical expansion (I suggest that the authors read this paper on chemical expansion: Phys. Chem. Chem. Phys., 2015, 17, 10028-10039) for the BCZTZICM sample. They claim that this is due to the hydration process. However, hydration process is known to be charge neutral (see this in any classic papers on proton conductors), i.e., no B-site cation redox involved. It is very strange to see that Co XANES show a peak shift upon switching to wet atmosphere (the peak

shift is very small though, maybe due to formation of a secondary phase in wet atmosphere).

Response to C2:

Thanks for reviewer's valuable insight and comments. We appreciate it for pointing this out. The contradiction between the charge neutrality principle in the classical hydration process and our XANES observation results you pointed out is really the core problem that we need to face and explain seriously. The reviewer's opinion has greatly helped us to improve the scientific rigor of the paper.

The reviewer's speculation that the small XANES peak shift might originate from secondary phase formation under wet atmosphere is a reasonable hypothesis. We have carefully verified this possibility through high temperature X-ray diffraction (XRD) analysis, which shows no evidence of any new phase formation. Therefore, we can conclude that the observed XANES changes are intrinsic responses of the BCZTZICM perovskite phase to the steam atmosphere.

We fully agree with the reviewer's core argument. The classical hydration reaction for proton incorporation:

This process is indeed charge-neutral and, in theory, should not involve a change in the valence state of the B-site cations. However, our reproducible XANES data clearly show a small but consistent shift of the Co K-edge to higher energy upon introducing steam at 600 °C. This shift typically indicates a decrease in local electron density around Co, suggesting a trend towards a higher apparent oxidation state.

This observation suggests that in the complex environment of a PCEC oxygen electrode, proton incorporation is not solely governed by the classical, charge-neutral hydration mechanism. Beyond that, proton absorption can proceed through alternative reaction pathways that *do* involve transition metal redox chemistry. Specifically, the following two mechanisms are well-established in the literature for oxygen electrodes:

The third mechanism explicitly involves a change in the valence of the redox-active transition

metal (e.g., Co, Mn, Fe, Ni). This pathway has been repeatedly demonstrated to contribute significantly to proton uptake in perovskite oxygen electrodes (Chem. Mater. 2019, 31, 8383-8393; Energy Environ. Mater. 2023, e12660; Nat. Catal. 2022, 5, 777-787). All of these mechanisms aim to form proton vacancies, thereby improving the proton conductivity of the electrode.

We acknowledge that using XANES alone to discuss this process is somewhat limited. In addition to the evidence mentioned above, we also characterized the material's physical adsorption capacity using H₂O-TPD and observed proton defect-related characteristic peaks via *in-situ* Fourier transform infrared spectroscopy. We believe that the combination of these multiple pieces of evidence supports the conclusion that BCZTZICM exhibits improved proton absorption capabilities.

We sincerely thank the reviewer for pointing this issue, which has significantly strengthened the scientific rigor of our manuscript.

Action: We have revised the manuscript and supplementary information as below:

Page 13, Line 250-253

The steam tolerance of BCZTZICM was evaluated by high-temperature XRD during cyclic transitions between dry and humid air atmospheres. As shown in Supplementary Fig. 23, the material exhibits only reversible lattice expansion/contraction, without emergence of secondary phases.

Page 2, Line 26

Moreover, the thermally driven mild oxygen release can be further offset by beneficial **proton uptake**, thereby increasing the thermomechanical durability of the oxygen electrode.

Page 2, Line 26

Furthermore, X-ray absorption near-edge structure (XANES) and *in-situ* Fourier transform infrared (FT-IR) spectroscopy confirms an enhanced **proton uptake** capability, which synergistically mitigates the microstructural degradation induced by electrode cracking.

Page 12, Line 233-236

Steam-treated sample shows elevated valences, indicating **proton uptake** partially offsets thermal expansion. **This process (Supplementary Note 2) generates protonic defects that induce the changes in**

local electronic structure and ion valence. OH^- formation enables partial assessment of the proton uptake capacity via valence variations. After normalization for the elemental composition, lattice oxygen content variations quantify proton uptake capacity (Fig. 3d). BCZTZICM exhibits an exceptional proton uptake capability under humid conditions, with a lattice oxygen content increase of 0.056, compared with only 0.042 for BSCF, when switching from dry to humidified environments at 600 °C.

[Supplementary information]

Supplementary Fig. 23.

High temperature XRD patterns of BCZTZICM at 600 °C during switching between dry and humid air atmospheres, and magnified XRD patterns of the selected 2 theta range of 29.55-30.45°.

Supplementary Note 2

The proton uptake has the beneficial effect of enhancing the absorption of protons by the electrode, thereby strengthening the proton transport capacity. In general, proton defects can be generated by hydration reactions with oxygen vacancies, as illustrated by the following formula:

Where O_O^{\times} represents lattice oxygen, $V_O^{\prime\prime}$ is oxygen vacancies, and OH_O^{\cdot} represents proton defects. Beyond classical hydration pathways, PCEC oxygen electrodes can generate protonic defects through transition metal redox-mediated routes.

Where h^{\cdot} represents electronic holes, $Mn(III)_{Mn}^{\times}$ is Mn^{3+} ion. The proposed mechanisms, extendable to Fe/Co/Ni systems based on their redox activity, have been experimentally validated in relevant perovskite electrodes.

Comment 3:

It is very difficult to understand the key contributors to the increased performance of the BCZTZICM sample without knowing some fundamental properties of the perovskite (i.e., oxygen non-stoichiometry, ionic/electronic conductivity). It is entirely possible that the better performance compared with BSCF is NOT from “ionic dispersion”. The authors might want to reflect how to prove this causality.

Response to C3:

We sincerely thank the reviewer for raising this critical point regarding the causality behind the enhanced performance of BCZTZICM. We agree that a deeper understanding of the fundamental properties is essential to substantiate our claim that the "ionic dispersion" strategy enhances performance by stabilizing the proton conduction pathway, rather than boosting oxygen exchange

kinetics. To address this, we have conducted a series of complementary experiments, the results of which are now included in the revised manuscript (Fig. 2f, Supplementary Fig. 25-26).

Based on TG coupled with room-temperature X-ray absorption near-edge structure (XANES) spectroscopy to determine elemental valence states, we have determined the temperature-dependent oxygen non-stoichiometry ratio (δ) for all samples (Fig. 2f). The results reveal distinct oxygen desorption behaviors: BSCF exhibits rapid oxygen loss at elevated temperatures, generating excessive oxygen vacancies. BCZTZICM and BCFZY demonstrate significantly milder oxygen release profiles.

Electronic conductivity measurements were performed and are presented in Fig. S. BSCF shows a characteristic non-monotonic temperature dependence, directly correlated with its severe oxygen loss. This is due to excessive oxygen vacancies annihilating electron holes, leading to decreased carrier concentration and consequently reduced electronic conductivity at high temperatures. In contrast, BCZTZICM and BCFZY demonstrate significantly milder oxygen release profiles. Notably, BCZTZICM maintains more stable oxygen content at operating temperatures, with near-linear temperature-dependent conductivity behavior indicative of thermally activated electron hopping as the dominant charge transport mechanism, supporting the suppression of excessive oxygen vacancy formation. BCZTZICM maintains stable oxygen content at elevated temperatures, preserving the integrity of proton diffusion pathways that depend on lattice oxygen. In contrast, BSCF undergoes excessive lattice oxygen release, and the resulting vacancies disrupt proton diffusion pathways, creating high-barrier diffusion sites.

To evaluate oxygen electrocatalytic activity, electrical conductivity relaxation (ECR) measurements were conducted to extract oxygen surface exchange coefficients (k_{chem}) and bulk diffusion coefficients (D_{chem}) (Fig. S). Importantly, BCZTZICM does not show significantly superior oxygen activity compared to BSCF under dry air conditions. BCZTZICM does not exhibit notably superior oxygen activity compared to BSCF. Symmetric cell measurements using GDC electrolytes further reveal that BCZTZICM shows slightly lower ORR activity than BSCF and BCFZY under dry air conditions. Therefore, the superior performance of BCZTZICM in PCECs is not due to enhanced oxygen reduction kinetics but is primarily attributed to enhanced proton uptake and optimized proton conduction capability, facilitated by a stabilized proton-conducting lattice. We thank the reviewer

again for prompting this more rigorous and nuanced discussion, which has significantly strengthened our manuscript.

Action: We have revised the manuscript and supplementary information as below:

Page 9, Line 177-180

Thermogravimetric analysis demonstrates a mass loss of 1.12% for BCZTZICM and higher losses for BSCF and BCFZY. Based on room-temperature XANES, the oxygen non-stoichiometric ratios were obtained. BCZTZICM demonstrates a higher concentration of inherent oxygen vacancies compared to BCFZY, which is beneficial to surface activity.

Fig. 2 Multielement micro-doping enhances structural homogenization. (a) Bond length distributions, (b) minima of the Electron localization function (ELF) along bond paths, and (c) O-ELF values for BCZTZICM, BSCF, and BCFZY, using standard deviation (σ) as the metric for internal uniformity. (d) Average O-ELF values, (e) oxygen vacancy formation energy, and (f) TG and oxygen non-stoichiometric ratio data of BCZTZICM BSCF and BCFZY samples.

Page 15, Line 304-314

Electronic conductivity measurements (Supplementary Fig. 25) reveal the characteristic of BSCF non-monotonic temperature dependence, directly correlated with severe oxygen loss. Excessive oxygen vacancies annihilate electron holes, reducing carrier concentration and conductivity at high

temperatures. In contrast, BCZTZICM maintains stable oxygen content at operating temperatures, exhibiting near-linear temperature-dependent conductivity dominated by thermally activated electron hopping. This behavior confirms suppressed oxygen vacancy formation, preserving proton diffusion pathways that depend on lattice oxygen. BSCF's excessive oxygen release disrupts these pathways, creating high-barrier diffusion sites.

Electrical conductivity relaxation (ECR) measurements were conducted to extract oxygen surface exchange coefficients (k_{chem}) and bulk diffusion coefficients (D_{chem}) (Supplementary Fig. 26). Notably, BCZTZICM shows no superior oxygen activity versus BSCF under dry air. Electrochemical impedance spectroscopy (EIS) of symmetric cells determines of electrode-specific contributions. Dry-air testing of $\text{Gd}_{0.2}\text{Ce}_{0.8}\text{O}_{1.9}$ (GDC) based cells eliminated the confounding effects of proton conduction.

Supplementary information

Supplementary Fig. 25.

Electronic conductivities of the BCZTZICM, BSCF, and BCFZY samples at 300 – 800 ° C in air, as measured by the 4-probe DC conductivity technique.

Supplementary Fig. 26.

The fitted values of D_{chem} and k_{chem} of BCZTZICM, BSCF, and BCFZY samples at temperature range of 550–700 ° C from electronic conductivity relaxation curves.

Point-to-Point Responses to Reviewers' Remarks and Suggestions

First, we express our sincere gratitude to editor for handling our manuscript and for the insightful feedback provided by the reviewers. Their comments have greatly improved the quality and clarity of the paper.

Below, the following are the point-by-point response to reviewers' comments. All changes in the manuscript have been marked in red for convenience.

Reviewer #3 (Remarks to the Author):

The authors have revised the manuscript and I have been asked by the editor to review whether my comments have been fully addressed. Unfortunately, I do not believe that the questions I raised have been fully answered. I still fail to see that the concept of “micro-doping”, which is claimed to induce a more “homogenous” distribution of oxygen vacancies, can really pan out. Still, the link between “micro-doping” to more homogenous oxygen vacancy distribution to better electrochemical performance cathode is still rather weak, in my opinion. Please note that I am only commenting on whether the authors have addressed my previous comments. I have no intention of further commenting on the quality of this work or introducing new questions.

Comment 1:

Regarding the more homogenous distribution of oxygen vacancies claimed by the authors:

The authors claimed that they reached this conclusion not by directly observing the distribution of oxygen vacancies in microscopic scales (due to technical difficulties). Rather, the conclusion is reached circumventively by looking at the distribution of cations. Firstly, I cannot see why the RDF analysis from the APT results show a more homogenous distribution of cations for BCZTZICM (in fact, the RDF of this sample show a higher fluctuation, indicating potential inhomogeneity). Secondly, even if the cations are randomly distributed, adding more dopant might introduce more electrostatic interactions between particular dopant and oxygen vacancies. The AP-XPS data is not very meaningful, in the sense that the Oads peak has multiple origins, ranging from surface -OH group to contaminants such as adventitious hydrocarbons to sulfuric groups. Since the BCZTZICM shows a persistent Oads peak even at 600 degree C, I suspect that most likely this peak is due to surface contamination of S/Si

species (see for example, Riedl et al., ACS Appl. Mater. Interfaces 2023, 15 (22), 26787–26798. <https://doi.org/10.1021/acsami.3c03952>). In any case, this O 1s peak CANNOT be linked to oxygen vacancies (see Wang et al., Journal of the European Ceramic Society 2024, 44 (15), 116709. <https://doi.org/10.1016/j.jeurceramsoc.2024.116709>).

Response to C1:

We sincerely appreciate the Reviewer's insightful comments regarding the RDF analysis and NAP-XPS interpretation.

Regarding the RDF analysis from APT, the observed fluctuations in normalized concentration for BCZTZICM (Supplementary Fig. 8) are expected. The key metric is the proximity of the normalized concentration profiles to unity across various radial distances, which indicates similar association or segregation tendency between different ion species. This contrasts with BSCF and BCFZY, whose RDF show clearer tendencies for like-ion association (e.g., Co-Co in BSCF) or exclusion (e.g., Co-Zr in BCFZY). The fluctuations in RDF of BCZTZICM thus reflect a more randomized, dispersed cationic distribution without long-range segregation, which is the intended outcome of our multi-element micro doping strategy.

We fully concur with the Reviewer's consideration that multi-dopants may introduce enhanced electrostatic interactions. Indeed, such strengthened electrostatic interactions would suppress proton-electron coupled migration. The increased average oxygen vacancy formation energy and the notably lower TEC for BCZTZICM are proof of the modified and homogenized chemical bonding environment, which includes these electrostatic interactions. This aligns with established literature where high-entropy or multi-doping strategies suppress lattice oxygen loss and reduce TEC.

Critically, our proton diffusion barrier calculations reveal that BCZTZICM does not possess the lowest individual hopping barriers. Specifically, the minimum proton hopping barrier between adjacent oxygen sites in BSCF can be as low as 0.20 eV, compared to 0.24 eV for BCFZY and 0.25 eV for BCZTZICM. However, the maximum barrier sites in BSCF reaches 0.87 eV, significantly exceeding those of BCFZY (0.51 eV) and BCZTZICM (0.41 eV). In proton conduction processes,

high-barrier sites effectively obstruct potential migration pathways, creating bottlenecks that limit overall proton conductivity regardless of the presence of low-barrier regions.

The strategy of suppressing electrostatic interactions to enhance proton conductivity has been extensively studied in classical proton conductor materials, and we fully agree with the Reviewer's perspective. However, for PCEC oxygen electrodes, several unique considerations apply: electrode materials must maintain substantial ORR/OER activity, necessitating the incorporation of redox-active transition metals (Co, Fe, etc.) as catalytic centers. This requirement inevitably introduces abundant oxygen vacancies and severe local valence and coordination heterogeneity, resulting in the coexistence of numerous low-barrier and high-barrier regions. We believe this one of the primary bottlenecks currently limiting the development of high-performance oxygen electrode materials.

Regarding the NAP-XPS measurements: We appreciate the Reviewer's concern about potential surface contamination. We wish to clarify that all samples were prepared following identical protocols within an argon-atmosphere glovebox, and during the in-situ heating process, high-purity bottled air (rather than ambient atmosphere) was continuously introduced, thereby minimizing the possibility of adventitious contamination.

Furthermore, evidence supporting enhanced metal-oxygen bonding in BCZTZICM is also derived from TGA and O₂-TPD. We have revised the manuscript to clarify these points and have removed part about the NAP-XPS O 1s spectra.

Action: We have revised manuscript as below.

Page 7, Line 143-153:

To gain deeper insights into the local chemical environment and potential nanoscale segregation tendencies, radial distribution function (RDF) analysis derived from APT was performed^{25,26}. The key metric is the proximity of the normalized concentration profiles to unity across various radial distances, which indicates similar association tendency between different ion species. As illustrated in Supplementary Fig. 8, with Co ions as the center, the normalized concentrations of high-abundance Co cations and oxygen anions approach unity, while trace-doped In³⁺ and Zr⁴⁺ ions display vibrant

concentration profiles within the analyzed region (constrained by limited abundance), collectively indicating a remarkably homogeneous spatial distribution of Co ions, O^{2-} , and dopant ions. In contrast, BSCF suggests a slight tendency for Co ions to disperse from Fe, implying a preferential association among like ions (Co-Co and Fe-Fe) within the analyzed region, and O^{2-} demonstrate a stronger association with Co than with Fe. For BCFZY, Co exhibits a preferential association with Y^{3+} , Fe ions and O^{2-} , while displaying a relative exclusion toward Zr^{4+} .

Page 8-9, Line 166-171:

While BSCF contains low-barrier sites (0.20 eV), high-barrier regions (0.87 eV) obstruct migration, limiting conductive pathways and the overall proton conductivity. Notably, BCZTZICM maintains exceptional uniformity, with only 0.16 eV difference between maximum (0.41 eV) and minimum (0.25 eV) barrier sites, eliminating bottleneck effects and creating extensive networks of facile transport pathways that enable rapid proton diffusion in three dimensions.

Page 10, Line 193-195:

Thermogravimetric analysis demonstrates a mass loss of 1.12% for BCZTZICM and higher losses for BSCF and BCFZY, indicating an enhanced metal-oxygen bond energy of BCZTZICM.

Comment 2:

2. Regarding the change of XANES energy shift upon hydration:

The authors explained the shift of XANES spectra by writing a “new” defect chemical reaction. However, I should note that the new defect chemical reaction is actually the hydrogenation process (i.e., splitting of H_2O into H_2 and O_2 , where H_2 reacts with perovskite to form reduced cations and protons). It is by no means hydration process. Unless the authors believes that their perovskite can induce spontaneous water splitting (and magically get rid of the oxygen gas molecules, too), this process cannot happen under water steam. I suggest that the authors review fundamental defect chemistry on proton conductors by consulting classic literature contributions (e.g., Merkle et al., *Annu. Rev. Mater. Res.* 2021. 51:461–93).

Response to C2:

We thank the reviewer for the insightful comment and for pointing us to the fundamental reference. The Reviewer correctly identifies the hydrogenation mechanism, wherein water dissociates into H₂ and O₂, with H₂ subsequently reducing the perovskite to form reduced cations and protonic defects. This process indeed consumes electron holes, leading to decreased cation valence states and, in materials with weak metal-oxygen bonding, potential cation exsolution (as recently demonstrated in ACS Energy Lett. 2025, 10, 4948–4956 by Woochul *et al.*).

However, the mechanism discussed in this manuscript is NOT hydrogenation. The experimental results show that the valence of transition metal cations INCREASES after introducing water vapor, rather than decrease of valence. This is derived from lowering of antibonding O 2*p* states that hybridize with transition metal 3*d* orbitals, coupled with the influence of incorporated protonic defects on the local coordination environment of transition metal ions.

This mechanism has been experimentally validated through *operando* X-ray absorption spectroscopy studies by Aoki and Zhou, respectively. (Chem. Mater. 2019, 31, 8383–8393, Energy Environ. Mater. 2023, 0, e12660) This demonstrated that proton incorporation can induce oxidation of B-site cations in specific perovskite systems. Similarly, in steam electrolysis via solid oxide electrolysis cells, water vapor can also induce re-oxidation of the Ni-YSZ hydrogen electrode to NiO-YSZ. Therefore, hydrogen is typically employed as a carrier gas to mitigate the re-oxidation of nickel.

It is important to note that this oxidative hydration mechanism is not universally observed across all perovskite compositions. For instance, Aoki *et al.* subsequently reported that La_{0.8}Ca_{0.2}Co_xNi_{1-x}O_{3-δ} does not exhibit this behavior (ACS Appl. Energy Mater. 2021, 4, 554–563). It depends on the electronic structure characteristics of the specific perovskite system, particularly the degree of metal-oxygen covalency, the position of the Fermi level relative to the oxygen 2*p* band, and the local coordination flexibility around B-site cations.

We have revised the manuscript to clarify that the observed valence increase is attributed to this proton-incorporation-induced electronic modulation, and we have removed the previous simplified defect reaction to avoid confusion.

Action: We have revised manuscripts and Supporting Information as below.

Page 11, line 227-234:

Under dry air conditions, BCZTZICM demonstrates minimal lattice oxygen content change (-0.075) compared with substantial losses for BSCF (-0.247) during heating to 600 °C (Fig. 3d), indicating lower thermal expansion, which were calculated using the equation in Supplementary Note 1. Notably, steam-treated samples exhibit elevated B-site cation valences. This oxidative response upon proton uptake originates from a distinct mechanism: the lowering of antibonding O 2p states hybridized with transition metal 3d orbitals, coupled with modifications to the local coordination environment induced by protonic defect incorporation (Supplementary Note 2). This mechanism has been experimentally validated through *operando* XAS studies on analogous perovskite systems, depending on the specific electronic structure characteristics (Supplementary Note 2). After normalization for elemental composition, the lattice oxygen content variations derived from XANES analysis quantify the proton uptake capacity (Fig. 3d). BCZTZICM exhibits an exceptional proton uptake capability under humid conditions, with a lattice oxygen content increase of 0.056, compared with only 0.042 for BSCF, when switching from dry to humidified environments at 600 °C.

Supplementary Information, Page 47:

Beyond classical hydration pathways, PCEC oxygen electrodes can generate protonic defects through transition metal oxide mediated routes^{32, 33, 34}.

Where h^{\cdot} represents electronic holes, $Mn(III)_{Mn}^x$ is Mn^{3+} ion. The proposed mechanisms, extendable to Fe/Co/Ni systems based on their redox activity. This mechanism is derived from lowering of antibonding O 2p states that hybridize with transition metal 3d orbitals, coupled with the influence of incorporated protonic defects on the local coordination environment of transition metal ions. It also has been experimentally validated in relevant perovskite electrodes. It should be noted that this oxidative proton uptake mechanism is not universally observed across all perovskite compositions, depending on the degree of metal-oxygen covalency, the position of the Fermi level relative to the oxygen 2p band, and the local coordination flexibility around B-site cations.

Comment 3:

3. Regarding the link between micro-doping and enhanced performance:

Based on the reasons above, as well as comments I noted in my previous review opinion, I still fail to see why the enhanced electrochemical performance HAS TO be linked with the concept of micro-doping and homogenous distribution of oxygen vacancies. In my opinion, the authors can provide clear information to (rather than mislead) the community by simply reporting that they have found a new cathode material for SOCs that can out-perform BSCF, as well as reporting the basic properties (lattices, ionic/electronic conductivities, hydration behaviors, etc.), rather than making a complicate story as shown in the current form of the manuscript.

Response to C3:

We sincerely thank the reviewer for reiterating this fundamental question. We agree that establishing a clear causal link is paramount. Our revisions and the data presented aim to demonstrate that the enhanced performance is not merely a coincidence but a direct consequence of the atomic-scale homogenization achieved specifically through our multi-element micro-doping strategy. The multi-element micro-doping strategy minimizes ionic association in BCZTZICM, as evidenced by APT/RDF and FT-IR. This atomic-scale homogenization, corroborated by theoretical calculations, helps eliminate high energy barrier bottlenecks for proton diffusion, and evidenced by DRT analysis of EIS. Consequently, while strengthened electrostatic interactions slightly reduce the intrinsic ORR activity, they collectively enable enhanced proton conductivity, superior proton uptake, and optimized thermomechanical stability.

Regarding the homogeneity of vacancy distribution, this was a speculation based on our current experimental and theoretical results. This content has been removed from the manuscript. We have revised the manuscript accordingly.

Page 10, Line 205-207:

Integration of APT findings with theoretical calculations reveals that materials exhibiting severe ionic localization (*e.g.*, BSCF) readily form excessive thermal stress ultimately inducing electrode cracking (Fig. 3a).

Page 10, line 215-216:

Multielement B-site doping in BCZTZICM effectively suppresses lattice oxygen loss through uniform local ion coordination environment, yielding near-linear behaviour.

Page 12, line 237-239:

The structural homogeneity ensures similar local atomic coordination, which leads to consistent lattice oxygen desorption across long-range domains.

Page 13, line 272-278:

BCZTZICM exhibits an exceptional proton uptake capability under humid conditions, with a lattice oxygen content increase of 0.056, compared with only 0.042 for BSCF, when switching from dry to humidified environments at 600 °C. The multi-element micro-doping, by maximizing ionic dispersion, creates this enhanced structural homogeneity, which in turn leads to a greater availability of uniform active sites, resulting in significantly improved proton uptake capability.

Page 16, line 320-322:

Electronic conductivity measurements (Supplementary Fig. 25) reveal the characteristic of BSCF non-monotonic temperature dependence, directly correlated with severe oxygen loss. Excessive oxygen vacancies annihilate electron holes, reducing carrier concentration and conductivity at high temperatures. In contrast, BCZTZICM maintains stable oxygen content at operating temperatures, exhibiting near-linear temperature-dependent conductivity dominated by thermally activated electron hopping. This demonstrate that lattice oxygen desorption in BCZTZICM exhibits a stochastic desorption characteristic. In contrast, the non-linear conductivity responses about temperature in BSCF and BCFZY are associated with the inhomogeneity of local structures.

Reviewer #4 (Remarks to the Author):

This manuscript reports a thoughtfully designed multielement micro-doping strategy to engineer a highly ionic-dispersed BaCoO₃-based oxygen electrode for reversible proton ceramic electrochemical

cells (R-PCECs). The authors convincingly demonstrate that atomic-scale compositional homogenization can simultaneously reduce proton diffusion barriers and improve thermomechanical stability under humid oxidizing conditions. Overall, the study is technically sound, well supported by experimental and computational evidence, and of clear interest to the solid-state electrochemistry and energy-materials communities. I am in favor of publication after revision, provided that the authors address the following points to further improve clarity and accessibility of the manuscript:

Comment 1:

1. Please clarify how the proposed multielement micro-doping strategy is fundamentally different from conventional high-entropy or random B-site doping in perovskites.

Response to C1:

We thank the reviewer for this important question.

High-entropy doping typically aims to achieve four characteristic effects: (1) the high-entropy stabilization effect favoring single-phase formation, (2) the lattice distortion effect arising from size mismatch, (3) the sluggish diffusion effect, and (4) the cocktail effect from synergistic elemental interactions. This typically dilutes the concentration of catalytically active B-site ions (*e.g.*, Co, Fe), which can compromise electrochemical activity.

Random B-site doping conventionally involves the incorporation of a single dopant element at moderate-to-high concentrations, which predominantly modifies the local coordination structure, often creating heterogeneous local strain fields.

Our multi-element micro-doping strategy is fundamentally distinct from conventional high-entropy or single-element doping in its design principle: instead of pursuing configurational entropy maximization or relying on the effect of an individual dopant, we employ a low total doping concentration (20% of B-sites) distributed across six distinct elements (Zr, Ti, Zn, In, Cu, Mo) at micro-doping levels (~3.3 % atom each). This approach prioritizes maximized ionic dispersion over entropy, which minimizes ion aggregation, disrupts extensive vacancy or strain clusters, and creates a homogeneous local bonding environment-while preserving a high bulk concentration (80%) of active Co ions to maintain electrocatalytic activity. The outcome is a synergistic effect where moderate lattice

strain enhances structural stability and atomic-scale homogenization facilitates uniform proton transport.

We have revised the manuscript to explicitly articulate these fundamental distinctions.

Page 5, lines 91-99:

This yielded the multi-element micro-doped BCZTZICM, which together with BSCF and BCFZY establishes a B-site elemental dispersion gradient design concept. Differs fundamentally from conventional high-entropy or single-dopant approaches. Instead of maximizing configurational entropy, multi-element micro-doping strategy distributes a low total doping concentration (20 % of B-sites) across six distinct elements (Zr, Ti, Zn, In, Cu, Mo) to prioritize ionic dispersion. This design suppresses ion aggregation and disrupts defect clusters while preserving 80 % active Co for high catalytic activity. The resulting homogeneous local environment, combined with moderate lattice strain from valence and ionic-radius diversity, enables uniform proton-transport pathways with low energy barriers and yields a low, near-linear thermal expansion. Ionic dispersion is the central mechanism, synergistically enhanced by valence diversity and controlled strain.

Comment 2:

2. Please explicitly state the core design principle (ionic dispersion vs. entropy vs. strain vs. valence diversity) responsible for the observed performance gains.

Response to C2:

The observed performance enhancement is principally driven by maximized ionic dispersion, achieved through multi-element micro-doping. This strategy intentionally utilizes valence diversity (Zn²⁺ to Mo⁶⁺) and a gradient of ionic radii (0.60-0.72 Å) to create a homogeneous, moderately strained lattice. While configurational entropy increases naturally, it is not the main mechanism. The core aim is to disrupt short-range ordering of ions and their associated defect clusters-especially Co/Fe aggregation and aligned oxygen-vacancy networks. This atomic-scale homogenization directly enables uniform proton-diffusion pathways with consistently low energy barriers and no severe bottlenecks, and suppressed anisotropic oxygen release, yielding a low, near-linear thermal expansion.

Thus, ionic dispersion is the central principle, with valence diversity and controlled strain serving as synergistic tools to achieve it, while preserving 80 % B-site Co to maintain high catalytic activity. The manuscript has been revised to clarify this design principle.

We have revised the manuscript as below:

Page 5, lines 91-99:

This yielded the multi-element micro-doped BCZTZICM, which together with BSCF and BCFZY establishes a B-site elemental dispersion gradient design concept. Differs fundamentally from conventional high-entropy or single-dopant approaches. Instead of maximizing configurational entropy, multi-element micro-doping strategy distributes a low total doping concentration (20 % of B-sites) across six distinct elements (Zr, Ti, Zn, In, Cu, Mo) to prioritize ionic dispersion. This design suppresses ion aggregation and disrupts defect clusters while preserving 80 % active Co for high catalytic activity. The resulting homogeneous local environment, combined with moderate lattice strain from valence and ionic-radius diversity, enables uniform proton-transport pathways with low energy barriers and yields a low, near-linear thermal expansion. Ionic dispersion is the central mechanism, synergistically enhanced by valence diversity and controlled strain.

Comment 3:

3. Suggest provide a quantitative definition of the proton-conduction bottleneck in oxygen electrodes, with representative literature values.

Response to C3:

We are grateful to the reviewer for valuable comment regarding the quantitative description of the proton conduction bottleneck.

The proton conduction bottleneck in oxygen electrodes represents a critical performance limitation in protonic ceramic electrochemical cells (PCECs). State-of-the-art electrolytes such as $\text{BaZr}_{0.8}\text{Y}_{0.2}\text{O}_{3-\delta}$ (BZY20) and $\text{BaZr}_{0.1}\text{Ce}_{0.7}\text{Y}_{0.1}\text{Yb}_{0.1}\text{O}_{3-\delta}$ (BZCYYb) exhibit proton conductivity of approximately $0.01\text{-}0.02\text{ S cm}^{-1}$ at $600\text{ }^{\circ}\text{C}$ under humidified conditions (Enrico Traversa *et al.* Nature Mater. 2010, 9, 846-852., Donglin Han *et al.* J. Mater. Chem. A, 2024, 12, 5875., J. Mater. Chem. A, 2018, 6, 18571). In contrast, conventional oxygen electrodes demonstrate significantly lower proton

conductivity values. The proton conductivity of conventional oxygen electrode materials, as evidenced by measurements utilizing diverse techniques-such as the bulk diffusion coefficient testing for (Ba/Sr)(Co/Fe/W)O_{3-δ}@PrBa_{0.5}Sr_{0.5}Co_{1.5}Fe_{0.5}O_{5+δ} (~10⁻⁵ S cm⁻¹ at 800 °C; Ling Zhao *et al.* Adv. Mater. 2024, 2405052), hydrogen permeation for La_{0.8}Sr_{0.2}Sc_{0.5}Fe_{0.5}O_{3-δ} (1.1×10⁻⁴ S cm⁻¹ at 600 °C; Lei Bi *et al.* SusMat. 2023, 3, 697-708.), and the Hebb-Wagner DC polarization method for BaGd_{0.3}La_{0.7}Co₂O_{6-δ} (4.4×10⁻⁵ S cm⁻¹ at 600 °C; Tadeusz Miruszewski *et al.* J. Mater. Chem. A, 2024, 12, 13488)-is consistently and significantly lower than that of state-of-the-art proton-conducting electrolytes. This dramatic conductivity mismatch (2-4 orders of magnitude) creates a severe transport bottleneck, confining electrochemical reactions to the narrow triple-phase boundaries (TPBs) where electrolyte, electrode, and gas phase meet. Consequently, the effective reaction zone is restricted to approximately 1-2 μm near the electrode-electrolyte interface, severely limiting electrode utilization and overall cell performance. This quantitative analysis underscores the critical need for developing triple-conducting (H⁺/O²⁻/e⁻) oxygen electrodes to extend the electrochemically active region.

We have revised manuscript as below:

Page 6, lines 111-116:

Proton conduction bottlenecks in oxygen electrodes critically limit PCEC performance. State-of-the-art electrolytes (BaZr_{0.8}Y_{0.2}O_{3-δ}, BZCYYb) exhibit proton conductivity of ~0.01-0.02 S cm⁻¹ at 600 °C under humidified conditions^{5, 19}. **The proton conductivity of conventional oxygen electrode materials, as evidenced by measurements utilizing diverse techniques-such as the bulk diffusion coefficient testing²⁰ for (Ba/Sr)(Co/Fe/W)O_{3-δ}@PrBa_{0.5}Sr_{0.5}Co_{1.5}Fe_{0.5}O_{5+δ} (~10⁻⁵ S cm⁻¹ at 800 °C), hydrogen permeation²¹ for La_{0.8}Sr_{0.2}Sc_{0.5}Fe_{0.5}O_{3-δ} (1.1×10⁻⁴ S cm⁻¹ at 600 °C), and the Hebb-Wagner DC polarization method²² for BaGd_{0.3}La_{0.7}Co₂O_{6-δ} (4.4×10⁻⁵ S cm⁻¹ at 600 °C) -is consistently and significantly lower than that of state-of-the-art proton-conducting electrolytes.** This severe conductivity mismatch confines electrochemical reactions to narrow triple-phase boundaries, restricting the effective reaction zone to approximately 1-2 μm near the electrode-electrolyte interface, limiting electrode utilization and overall cell performance.

Comment 4:

4. Suggest to clearly distinguish bulk proton transport from surface or interfacial proton transport in the discussion.

Response to C4:

We sincerely thank the reviewer for raising this important distinction. Bulk proton conduction in perovskite-based oxygen electrodes is predominantly governed by the Grotthuss mechanism in both fuel cell and electrolysis operational modes, involving thermally activated proton hopping between adjacent lattice oxygen sites via sequential O-H bond cleavage and formation. In contrast, surface and interfacial proton transport originates from the proton uptake process, wherein H₂O dissociated to form protonic defects (OH⁻). The source of protons differs fundamentally between the two operational modes: in electrolysis mode, protons are derived exclusively from H₂O dissociation at the oxygen electrode surface; whereas in fuel cell mode, the water vapor generated as a reaction product at the oxygen electrode undergoes subsequent proton uptake, thereby augmenting the protonic conductivity of the electrode. We have revised the manuscript to explicitly clarify these mechanism distinctions between bulk and surface proton transport processes.

We have revised manuscript as below:

Page 6, lines 122-128:

Proton conduction in perovskite oxide mechanisms exhibit temperature-dependent dominance. At

e

l

e

v

a

t

e

^

Page 14, lines 280-289:

Proton transport in perovskite-based oxygen electrodes follows the Grotthuss mechanism in both operational modes. Surface proton transport, in contrast, originates from the proton uptake process through H₂O dissociative chemisorption to form protonic defects. Fundamental distinctions arise in proton sources between operational modes: in fuel cell mode, protons originate from hydrogen dissociation at the hydrogen electrode, transporting through electrolyte, and forming water at the

r

a

t

oxygen electrodes, while water vapor generated as reaction product undergoes subsequent proton uptake, augmenting electrode protonic conductivity. Electrolysis mode generates protons exclusively through water dissociation at the oxygen electrode, with subsequent transport to the hydrogen electrode. The overall efficiency critically depends on electrode ability to rapidly adsorb water vapor, facilitate dissociative chemisorption, and transport protons into the bulk.

Comment 5:

5. Suggest to discuss whether multiple proton transport mechanisms may coexist under different temperatures or humidity conditions.

Response to C5:

We appreciate the Reviewer for raising this insightful point regarding temperature-dependent proton transport mechanisms.

Indeed, multiple proton transport mechanisms can coexist and exhibit temperature-dependent dominance in perovskite-based oxygen electrodes. At high temperatures (> 600 °C), the accelerated reaction kinetics allow proton migration via OH^- transport through oxygen vacancy sites, where the entire OH^- migrates as a mobile species, which is commonly referred to as the vehicle mechanism. However, as the operating temperature decreases, the activation energy associated with OH^- vehicular transport becomes prohibitively high (> 1 eV), rendering this pathway kinetically unfavorable.

Consequently, at intermediate-to-low temperatures characteristic of PCEC operation (450–600 °C), proton transport predominantly occurs via the Grotthuss (hopping) mechanism, wherein protons (H^+) form bonds with lattice oxygen atoms and migrate through sequential bond cleavage and reformation between adjacent oxygen sites. This proton hopping mechanism is supported by multiple lines of evidence, including isotope effect studies (Yoshihiro Yamazaki *et al.* *Nat. Mater.*, 2025. <https://doi.org/10.1038/s41563-025-02311-w>), electrochemical impedance spectroscopy (Zongping Shao *et al.* *Applied Catalysis B: Environmental*, 2023, 331, 122682), and temperature-dependent conductivity measurements (K.D Kreuer, *Solid State Ionics*, 1999, 125, 285-302., Yashima M., *Nat. Commun.* 2023 14, 7466.). It has also been directly observed in similar perovskite systems via techniques such as quasi-elastic neutron scattering (QENS) and neutron diffraction. Crucially, the characteristic activation energies we observe (typically 0.4–0.6 eV) align with the proton hopping

mechanism, and are distinct from the higher barriers (>1 eV) typically associated with hydroxyl group migration.

We have revised the manuscript to incorporate a discussion of these temperature-dependent mechanistic considerations.

Page 6, lines 122-128:

Proton conduction in perovskite oxide mechanisms exhibit temperature-dependent dominance. At

e
l
e
v
a
t
e

d
Comment 6:

6. Clarify how proton uptake affects oxygen vacancy concentration and lattice stability.

Response to C6:

m We thank the Reviewer for this question regarding the interplay between proton uptake, oxygen
p vacancy concentration, and lattice stability.

e
r At high temperatures, perovskite-based oxygen electrodes undergo substantial oxygen desorption,
g generating oxygen vacancies that serve as catalytically active sites for surface exchange reactions.
a However, this extensive vacancy formation simultaneously places the lattice in a thermodynamically
t metastable state, rendering it susceptible to structural degradation.

u
r Proton uptake through the formation of protonic defects that occupy oxygen vacancy sites. This
p process effectively reduces the oxygen vacancy concentration and, more critically, suppresses the
p pronounced valence changes of transition metal cations. Besides, the presence of protonic defects
partially compensated for the lattice contraction upon cooling associated with oxygen during thermal
(cycling. This is aligned with the “hydration-induced chemical expansion offset” concept (Xie *et al.*

>

6

Nature Communications, 2025, 16, 3154). We have revised the manuscript to explicitly discuss this mechanistic framework.

Page 13-14, lines 274-278:

Enhanced thermomechanical stability originates from two synergistic mechanisms: the homogeneous ionic distribution enables uniform oxygen release, yielding a reduced and near-linear thermal expansion coefficient; concurrently, proton uptake capability suppresses Co reduction while the accompanying chemical expansion compensates for thermal contraction during cooling.

Comment 7:

7. Please clearly label all APT elemental maps and color scales and discuss dataset representativeness.

Response to C7:

Thanks to the reviewers for pointing this out. The needle-shaped specimens for APT analysis were prepared from dense bulk material using focused ion beam milling, thereby mitigating potential surface-related artifacts. Each APT dataset captures over one million atoms, with a minimum analyzed volume exceeding $20 \times 20 \times 200 \text{ nm}^3$. This is far exceeding the unit-cell dimensions, ensuring statistically meaningful analysis of ionic distributions and enabling the detection of possible nanoscale segregation. Furthermore, the compositional homogeneity inferred from APT is corroborated by FT-IR (Supplementary Fig. 4). The proton diffusion energy barriers deduced from the structural analysis align consistently with results from electrochemical impedance spectroscopy and DFT calculations. The rest of the APT element distribution shown in Fig. 1g is fully shown in Supplementary Fig. 7.

We have made the following changes to the pictures of the manuscript:

Page 8:

Fig. 1 Dispersed B-site doping eliminates proton diffusion bottlenecks in BCZTZICM perovskite electrodes. (a) Refined XRD profiles of the prepared BCZTZICM. (b) HR-TEM image of BCZTZICM along (011), (120), and (111) plane. Structure diagram of PCEC, the amplification part is a brief sketch of the ion arrangement of (c) dispersed and (d) aggregated at B site of perovskite-type oxygen electrode. Schematic illustrating (e) abundant proton conduction pathways facilitated by discrete oxygen vacancies enabled through homogeneous ionic arrangement, and (f) regions of elevated proton diffusion barrier due to contiguous vacancies. (g) 3D reconstruction of the APT data showing mapping of the BCZTZICM sample with O distributions. Integrated line profiles show chemical composition alongside the z-axis of the analyzed needle. (h) Proton diffusion energies of BCZTZICM, BSCF, and BCFZY in the bulk.

Supplementary Information, Page 6-8

Supplementary Fig. 5.

3D reconstruction of the APT data showing mapping of the BSCF sample with Ba, Sr, Co, Fe, and O ionic distributions. Integrated line profiles show chemical composition at the (a) apex and (b) midsection regions of the analyzed needle.

Supplementary Fig. 6.

3D reconstruction of the APT data showing mapping of the BCFZY sample with Ba, Co, Fe, Zr, Y, and O ionic distributions. Integrated line profiles show chemical composition at the apex regions of the analyzed needle.

Supplementary Fig. 7.

3D reconstruction of the APT data showing mapping of the BCZTZICM sample with elements distributions. Integrated line profiles show chemical composition alongside the y-axis of the analyzed needle.

Comment 8:

8. Please provide key DFT modeling details (supercell size, vacancy configuration, proton concentration).

Response to C8:

We thank the reviewer for this valuable suggestion. The key parameters and their derivations are provided below.

Supercell Size

All density functional theory calculations were performed using a $3 \times 3 \times 3$ supercell constructed from the cubic perovskite unit cell, containing 27 ABO_3 formula units. The optimized supercell lattice parameters are $a=b=11.8720 \text{ \AA}$ and $c=25.8934 \text{ \AA}$ ($\alpha=\beta=\gamma=90^\circ$). For the stoichiometric perovskite structure, each ABO_3 unit contains 5 atoms, comprising 1 A-site cation, 1 B-site cation, and 3 oxygen atoms. Therefore, the total number of atoms in the stoichiometric supercell is $27 \times 5 = 135$ atoms, which

includes 27 A-site cations (Ba or Sr/Ba mixture), 27 B-site cations (Co-based with various dopants), and 81 oxygen atoms.

Oxygen Vacancy Configuration

To model the oxygen-deficient perovskite structure under realistic operating conditions, one oxygen vacancy was introduced at a specific lattice site within the supercell. The oxygen vacancy concentration δ can be calculated by dividing the number of vacancies by the number of formula units. Since we have 1 oxygen vacancy distributed among 27 formula units, the vacancy concentration is $\delta = 1/27 \approx 0.037$ per formula unit. Alternatively, considering that the supercell contains 81 oxygen sites in total, the percentage of vacant oxygen sites is $1/81 \times 100\% \approx 1.23\%$. This vacancy concentration was chosen to represent a dilute defect limit while maintaining computational efficiency.

Proton Concentration

For proton diffusion calculations, one hydrogen atom was incorporated into the oxygen-deficient supercell to simulate the proton-conducting state of the material. The proton concentration x per formula unit is calculated as the ratio of hydrogen atoms to the number of formula units, yielding $x = 1/27 \approx 0.037$ per formula unit. To express this in terms of atomic percentage, we note that after introducing one oxygen vacancy and one proton, the total number of atoms in the supercell becomes $135 - 1 + 1 = 135$ atoms. The hydrogen atomic percentage is therefore $1/135 \times 100\% \approx 0.74$ atom %.

The sites of calculated vacancy formation energies are presented in Supplementary Fig. 9, while the computational structures and energies related to oxygen vacancy formation energy, hydration and proton diffusion barriers are provided in the Supplementary Note 3. We have revised the manuscript as follows:

Page 26, lines 608-611:

Core-valence electron interactions were described by the projector-augmented wave (PAW) pseudopotentials^{54, 55}, and the electronic wavefunctions were expanded in a plane wave basis set with a kinetic energy cutoff of 500 eV. **The perovskite structures were constructed with a 3×3×3 supercell containing 27 ABO₃ formula units, and lattice parameters of 11.8720 Å×11.8720 Å×25.8934 Å ($\alpha=\beta=\gamma=90^\circ$). All model coordinates related to calculation are shown in Supplementary note 3.**

Electronic convergence was achieved when the energy difference between successive iterations was less than 10^{-5} eV, while geometric optimization was considered complete when the forces on individual atoms were below 0.05 eV/Å.

Comment 9:

9. Clarify whether reduced polarization resistance is limited by proton transport, oxygen exchange, or coupled kinetics.

Response to C9:

Thank reviewer for this important question. Our systematic kinetic analyses clarify that the notably reduced polarization resistance of the BCZTZICM electrode originates primarily from its significantly enhanced proton conduction. The DRT analysis of symmetric cells based on BZCYYb (Fig. 4e) shows a substantially weakened relaxation in the mid-frequency range, corresponding to a major improvement in bulk proton migration consistent with the low proton diffusion barriers revealed by DFT calculations. The weaker low-frequency response reflects enhanced surface proton uptake, aligning with the *in-situ* FT-IR and H₂O-TPD results. Concurrently, symmetric cell tests based on GDC indicate its ORR activity is adequate for high performance, and electrical conductivity relaxation measurements verify a surface oxygen exchange coefficient slightly inferior to that of BSCF, and better than BCFZY. Thus, the excellent overall performance stems from the synergy between enhanced bulk proton conduction, improved surface proton uptake, and sufficient, stable intrinsic oxygen exchange activity. We have revised manuscript as below:

Page 17, lines 346-349:

As shown in Fig. 4e, BCZTZICM exhibits substantially reduced contributions across all frequency ranges, with dramatic improvements in the mid-frequency regions corresponding to bulk proton transport. The weaker low-frequency response reflects enhanced surface proton uptake, aligning with the *in-situ* FT-IR and H₂O-TPD results. This confirms atomic-scale compositional homogenization creates extensive facile proton transport pathways optimizes proton uptake, and retains sufficient intrinsic oxygen activity. Comparison with state-of-the-art electrode materials

confirm the exceptional performance of BCZTZICM (Fig. 4f). ASRs for BCZTZICM at 600 °C (0.163 Ω cm²) substantially exceed advanced electrode metrics, establishing BCZTZICM as one of the highest-performing materials currently available^{9, 20, 27, 29, 30, 31, 32, 33, 34, 35, 36, 37, 38}.

Comment 10:

10. Is that possible to quantify and compare crack width/density among electrodes to support mechanical stability claims.

Response to C10:

We thank the reviewer for this suggestion. As recommended, we have now included comparison to support the claims of superior mechanical stability. The thermal cycling conditions and the corresponding crack widths observed in post-test microstructural analysis for reference electrodes are summarized in **Supplementary Table 2**. This data provides direct, comparative evidence for the enhanced thermomechanical durability of BCZTZICM oxygen electrode.

Page 13, lines 272-274:

The thermal cycling conditions and the corresponding crack widths observed in post-test microstructural analysis for reference electrodes are summarized in Supplementary Table 2.

Supplementary Information, Page 42:

Supplementary table 2. TEC of oxygen electrodes and crack width of electrode after thermal cycling.

Oxygen electrode materials	TEC ($10^{-6} K^{-1}$)	Thermal cycle conditions			Crack width (μm)	References
		Heating rate ($^{\circ}C$)	C	Cycle number		
SNC (based on SDC symmetric cell)	20.5	30	7.5	40	~1/0.6	
BSCF (based on SDC	22.4	30	6	40	~0.9	

symmetric cell)						
BCC						
(based on	25.89	10	6.7	35	~0.78/0.6	
BZCYYb						
symmetric cell)						
BCFZY						
(based on	22.56	20	10	40	1.38/1.54	This work
BZCYYb						
symmetric cell)						
BSCF						
(based on	26.08	20	10	40	1.89	This work
BZCYYb						
symmetric cell)						
SF						
(based on	30	30	10	35	~0.82/0.75	
BZCYYb						
symmetric cell)						
PBCF						
(based on	18.5	1.6	1.6	5	~4.2	
BZCYYb		10	5	15		
single cell)		20	10	50		

Comment 11:

11. Please explain whether improved thermomechanical stability arises mainly from suppressed oxygen release or proton uptake.

Response to C11:

Thank you for this valuable insight and comments. The improved thermomechanical stability stems from two synergistic mechanisms. Primarily, the significantly reduced thermal expansion

coefficient (TEC) and its near-linear temperature dependence originate from a more homogeneous oxygen release, enabled by the homogeneous ionic distribution. Secondly, the excellent hydration capability allows for pronounced proton uptake at elevated temperatures. This process not only suppresses the thermal reduction of Co ions but also utilizes the associated chemical expansion to offset the drastic thermal contraction during cooling cycles.

Page 13-14, lines 274-278:

Enhanced thermomechanical stability originates from two synergistic mechanisms: the homogeneous ionic distribution enables uniform oxygen release, yielding a reduced and near-linear thermal expansion coefficient; concurrently, proton uptake capability suppresses Co reduction while the accompanying chemical expansion compensates for thermal contraction during cooling.

Comment 12:

12. Please discuss dominant long-term degradation mechanisms observed or anticipated beyond 780 h.

Response to C12:

Thank you for raising this critical point. While the PCEC in this manuscript demonstrated excellent stability over 780 hours at 550 °C with low degradation rates ($<20 \mu\text{V h}^{-1}$), identifying and mitigating potential long-term degradation mechanisms is indeed crucial for achieving the thousands of hours of operation required for commercialization.

Based on our experimental observations and analysis, beyond the intrinsic chemical and microstructural stability of the materials, a key long-term failure risk originates from the silver paste typically used for sealing during button-cell testing. Under prolonged exposure to high temperature ($>500 \text{ }^\circ\text{C}$) and humid atmospheres, silver paste is susceptible to oxidation, creep, thermal stress mismatch, and mechanical degradation, leading to gradual seal failure. This can introduce additional gas leakage and crossover, not only increasing the ohmic resistance but also potentially triggering irreversible redox reactions at the electrodes, which could become the dominant factor in accelerated performance decay. Therefore, in future research and device development, we will prioritize optimizing the cell sealing strategy: moving away from the susceptible silver paste toward more promising approaches, such as high-temperature ceramic-based sealants or compressed mica gasket

seals. Concurrently, we will continue to monitor the phase stability of the hydrogen electrode and the surface evolution of the oxygen electrode under polarization. Through these systematic engineering and material optimizations, we aim to achieve even lower degradation rates, meeting the stringent requirement for extended operational lifetime in PCEC.

Page 19, lines 393-395:

The long-term galvanostatic testing at 550 °C (Fig. 5f) demonstrates stable performance over 780 hours, with degradation rates of 19.3 and 16.9 $\mu\text{V h}^{-1}$ in fuel cell mode and electrolysis mode, respectively. **The degradation originates from seal failure induced by silver sealant oxidation, creep, thermal stress mismatch, and mechanical deterioration under high-temperature and humid atmospheres.** Recent studies on oxygen electrode materials and corresponding long-term stability in PCEC summarized in Supplementary Table 5^{47, 48}.

Comment 13:

13. Please strengthen comparison with state-of-the-art oxygen electrodes using normalized metrics where possible.

Response to C13:

We thank the reviewer for raising this point. We have added a comparison of polarization resistance to the single-cell performance section, providing a more direct and standardized assessment of the electrode performance.

Page 19, lines 382-384:

Performance **and impedance** comparison establishes BCZTZICM among higher-performing PCEC systems (Fig. 5d **and Supplementary Table 3**)^{4, 20, 27, 28, 30, 37, 38, 39, 40, 41, 42, 43, 44, 45, 46}.

Supplementary Information, Page 43:

Supplementary Table 3. Performance and impedance comparison of oxygen electrodes in fuel cell and electrolysis modes at 600 °C.

Oxygen electrode	Peak power density (mW cm^{-2})	Polarization resistance at OCV ($\Omega \text{ cm}^2$)	Current density at 1.3 V (mA cm^{-2})	References
------------------	---	--	---	------------

CBSLCC	1660	0.112	1760	
PBSLCC	1160	0.112	1750	
BCZTZICM	1560	0.11	2000	This work
PLNBSCC	1210	0.17	1950	
BCT20 with PLD layer	1640	0.12	1600	
BSC-PBSCF	1640	0.04	/	
PBCHf10	1490	0.07	2780	
BCCY	743	0.113	/	
PNMCFC-PBC	1720	0.08	2170	
N-XFN	790	0.167	1700	
BCFN	1207	0.197	1511	
C/H-BSCF	1670	0.157	1230	
BSCF-RC _x	1100	0.13	1490	
NAUP-PNC	1501	0.07	1316	
PBN-BCZYYb CCS	1160	0.25	1460	

Comment 14:

14. I am curious whether similar performance could be achieved with fewer dopant elements.

We thank the Reviewer for this insightful question. As discussed in the manuscript, multielement synergy is essential for achieving ionic dispersion, strain balance, and valence modulation, objectives that cannot be readily accomplished with fewer dopant elements, as exemplified by BCFZY and BSCF.

To conceptualize this, we calculated the configurational entropy as a function of dopant species number at a fixed 20% B-site dopant concentration with 80% cobalt content (see figure below). The results reveal that configurational entropy increases progressively but with diminishing returns as more dopant species are introduced, suggesting that structural stabilization is achieved beyond a certain

compositional complexity. However, excessive doping would compromise electrocatalytic activity due to over-enhanced lattice distortion and excessively strengthened metal-oxygen bonding that reduces lattice flexibility.

To address the Reviewer's question experimentally, we synthesized single-element doped $\text{BaCo}_{0.8}\text{X}_{0.2}\text{O}_{3-\delta}$ ($\text{X}=\text{Zr}, \text{Ti}, \text{Zn}, \text{In}, \text{Cu}, \text{Mo}$) compositions. Unfortunately, none of these exhibited the desired cubic perovskite structure. We also synthesized a quaternary-doped cubic perovskite, $\text{BaCo}_{0.8}(\text{Zr}_{1/4}\text{Ti}_{1/4}\text{Cu}_{1/4}\text{Mo}_{1/4})_{0.2}\text{O}_{3-\delta}$ (BCZTCM), and evaluated its electrochemical performance using symmetric cells based on both GDC and BZCYYb electrolytes (see figure below). The results demonstrate that BCZTZICM with six micro-dopants exhibits superior performance compared to the quaternary-doped analogue.

In future work, we will continue to refine this design concept by systematically varying the active element content to develop further optimized oxygen electrodes.